# HOX gene expression in the developing human spine

John E. G. Lawrence[1,2], Kenny Roberts[1], Elizabeth Tuck[1], Tong Li[1], Lira Mamanova[1], Petra Balogh[3], Inga Usher[4], Alice Piapi[4], Pavel Mazin[1], Nathaniel D. Anderson[1], Liam Bolt[1], Laura Richardson[1], Elena Prigmore[1], Xiaoling He[5,6], Roger A. Barker[5,6], Adrienne Flanagan[7], Matthew D. Young[1], Sarah A. Teichmann[1], Omer Bayraktar[1,9] ✉ & Sam Behjati[1,8,9] ✉

Positional coding along the anterior-posterior axis is regulated by *HOX* genes, whose 3' to 5' expression correlates with location along this axis. The precise utilisation of *HOX* genes in different human cell types is not fully understood. Here, we use single-cell and spatial-transcriptomics, along with in-situ sequencing, to create a developmental atlas of the human fetal spine. We analyse *HOX* gene expression across cell types during development, finding that neural-crest derivatives unexpectedly retain the anatomical *HOX* code of their origin while also adopting the code of their destination. This trend is confirmed across multiple organs. In the axial plane of the spinal cord, we find distinct patterns in the ventral and dorsal domains, providing insights into motor pool organisation and loss of collinearity in *HOXB* genes. Our findings shed new light on *HOX* gene expression in the developing spine, highlighting a *HOX* gene 'source code' in neural-crest cell derivatives.

The first sign of anatomical patterning in human embryos is the emergence of the primitive streak, which establishes the anteroposterior (rostrocaudal) and left-right axes[1]. Molecularly, positional coding of this axis is underpinned by segment-specific expression of *HOX* genes, a group of transcription factors that contain a common DNA-binding homeodomain sequence[2]. Spread over four chromosomes, *HOX* genes are grouped into four clusters (A, B, C, and D). During gastrulation, these genes are expressed in a manner dictated by their position within the cluster, such that 3' genes contribute to rostral identity, and 5' to caudal; a pattern first described in *Drosophila*[3]. In vertebrates, *HOX* genes are activated sequentially in a temporally-restricted manner, beginning with 3' transcription, with the timing of progression through the cluster governed by CTCF binding sites[4–6].

Although the number of *HOX* genes and clusters varies across species, this collinearity is remarkably conserved[7–10]. In addition to their ability to dictate cell fate based on its position in the embryo, *HOX* genes also function during development to govern differentiation irrespective of a cell's rostrocaudal position[11,12].

The positional identity conferred by *HOX* genes at this early stage is thought to be highly conserved through development, as exemplified by somite transplantation experiments in the chick and retinoic-acid-induced posteriorisation of the mouse vertebral column[13,14]. Decades of work with such model organisms has thoroughly characterised *HOX* gene expression patterns in vertebrates, however subtle differences between these organisms' *HOX* genes and those of humans do exist[15]. Consequently, the precise utilisation of the rostrocaudal

[1]Wellcome Sanger Institute, Wellcome Genome Campus, Hinxton CB10 1SA, UK. [2]Department of Trauma and Orthopaedics, Cambridge University Hospitals NHS Foundation Trust, Addenbrooke's Hospital, Box 37Hills Road, Cambridge CB2 0QQ, UK. [3]Department of Cellular and Molecular Pathology, Royal National Orthopaedic Hospital, Brockley Hill, Stanmore HA7 4LP, UK. [4]University College London Great Ormond Street Institute of Child Health, London, UK. [5]Department of Clinical Neurosciences, University of Cambridge, CB2 0QQ Cambridge, UK. [6]Wellcome-MRC Cambridge Stem Cell Institute, Jeffrey Cheah Biomedical Centre Cambridge Biomedical Campus, Puddicombe Way, Cambridge CB2 0AW, UK. [7]Research, Department of Pathology, University College London (UCL) Cancer Institute, London WC1E 6DD, UK. [8]Department of Paediatrics, University of Cambridge, Cambridge CB2 0QQ, UK. [9]These authors jointly supervised this work: Omer Bayraktar, Sam Behjati. ✉e-mail: ob5@sanger.ac.uk; sb31@sanger.ac.uk

*HOX* code across human fetal cell types remains incompletely described.

Here, we set out to examine *HOX* gene expression directly in the human developing spine by forming a highly detailed tissue atlas along the rostrocaudal (antero-porterior) axis using three complementary high-resolution mRNA assays; single-cell RNA sequencing (scRNAseq), Visium spatial transcriptomics (ST) and Caratana in-situ sequencing (ISS). Our aim was to delineate the expression of *HOX* genes along the rostro-caudal axis across all captured cell types at high resolution, whilst disentangling their roles of patterning and differentiation within the context of the developing spine. We show varying *HOX* gene expression across cell types through development, highlighting that cells derived from the neural crest express *HOX* genes that correspond to their point of origin along the rostrocaudal axis. We validate this finding in the fetal limb, gut and adrenal gland using scRNAseq and use ISS to show neurons fall into distinct categories with regards *HOX* expression. Using ST and ISS, we shed new light on *HOX* expression across the dorsoventral axis in the developing spinal cord along its rostrocaudal length.

## Results

### A developing human spine atlas resolved in time and space

We first defined the cellular landscape of the human developing spine. We collected 7 spines from fetuses aged between 5 and 13 weeks post-conception. From post-conception week (PCW) 9 onwards (*n* = 5), we dissected each spine into precise anatomical segments along the rostrocaudal axis using anatomical landmarks and processed these segments separately (Fig. 1A; see Methods). This enabled us to delineate the inherent rostrocaudal maturation gradient of the fetal spine – the temporal maturation difference of approximately 6 hours that exists between each vertebral level during development[16]. We processed fresh tissues using standard techniques to generate single-cell suspensions, enriched for viable cells, from which we derived single-cell mRNA libraries using a droplet-based method (Chromium 10X; see Methods). Applying widely adopted quality filters (see Methods), we obtained count tables of transcripts of approximately 174,000 cells. These segregated into 61 distinct cell clusters (Supplementary Fig. 1A–D; Fig. 2A & B; Supplementary Data 1 for QC), broadly representing neuro-glial (*n* = 30,376), mesenchymal progenitors (n = 15,756), osteochondral (*n* = 43,171), muscle (*n* = 27,128), fibrous (*n* = 19,080), tendon (*n* = 20,445), meningeal (*n* = 3521), dermal (*n* = 552), haematopoetic (*n* = 9214), and endothelial (*n* = 4763) cells. We also captured rare cell types, such as hypertrophic chondrocytes (*n* = 16), notochordal cells (*n* = 33), cord floor (*n* = 38) and roof (*n* = 83) plate cells (Figs. 1B; 2A, B; Supplementary Data 2 for marker genes).

To validate and spatially resolve this census of cell types at PCW7 and PCW9, we utilised the Visium assay (Chromium), which provides 50µm resolution readouts of gene expression. We sectioned spines axially at different anatomical levels and examined each slide, applying the cell2location algorithm to obtain estimated cell type abundancy values for each voxel (Supplementary Fig. 1E for reconstruction accuracy)[17]. At PCW7, we also applied a 123-gene Cartana in-situ sequencing protocol (Chromium) to axial sections (see Methods). Consecutive axial sections were used for the two assays in order to maximise biological similarity. We were thus able to place populations of cells into their anatomical context, validated by the expression of classical marker genes (Fig. 1C–E). At PCW7, we were able to map both neuroprogenitors (NP) and post-mitotic cord neurons, together with marker genes, along the dorsoventral axis of the nascent cord at single-cell resolution (Fig. 2A–E). In doing so, we support the previous finding that *PAX7* is expressed by the floor plate in humans, and capture a bridging population (Differentiating Neuroprogenitors) between progenitors and neurons that express *NEUROD4*; a molecule implicated in neurogenesis following cord injury (Fig. 2B, D)[18,19]. At PCW9, mapping cells of the peripheral nervous system revealed clear segregation of

transcriptionally similar clusters of glia to either the nerve root emerging from the cord (Schwann cell 1), the dorsal root ganglion (satellite cell) or the exiting spinal nerve tract (Schwann cell 2), whilst mapping sensory and sympathetic neuronal cells to anatomically appropriate locations (Figs. 1E; 3A–C). Examining heterogeneity over time, the fraction of cell states in each compartment changed through development (Fig. 3D). In the mesenchymal compartment, multipotent progenitors dominated early samples, and were gradually replaced by more differentiated cell types such as chondrocytes, osteoblasts, tenocytes and fibroblasts(Fig. 3D & E). Similarly, the neural crest compartment at PCW5 and 7 was dominated by sensory neuron progenitors and immature schwann cells, with more differentiated neurons and glia emerging thereafter (Fig. 3D, E). Similarly, progenitors accounted for the majority of cell types between PCW5 and 9, with CNS glia dominating thereafter (Fig. 3D & E). Our developmental atlas can be freely accessed at https://spinal-development.cellgeni.sanger.ac.uk/.

### Rostrocaudal *HOX* expression in stationary cell types

Turning to our main question, we examined the pattern of *HOX* gene expression across different cell types. For this, we first excluded the early-stage fetuses, which could not reliably be dissected into precise anatomical segments. For the initial analyses, we categorised cell types into anatomically mobile (haematopoetic cells), neural crest-derived cells or stationary (all other cell types; Fig. 1B) We first aimed to identify *HOX* genes that reliably represented positional information across all stationary cells, regardless of type. We grouped stationary cell types and performed differential expression testing by region using the Willcoxon rank-sum test corrected for multiple comparisons to determine statistically significant trends in rostrocaudal *HOX* gene expression. In order to refine this positional code, we next examined the expression of the genes which reached significance in each of the anatomically stationary cell types separately and excluded any genes that exhibited expression patterns peculiar to a single cell type, presumably due to their playing a tissue-specific (and hence position agnostic) role in development during the period sampled. For example, *HOXA6*, *HOXD3*, *HOXD4* and *HOXD8* were expressed ubiquitously by tendon cells across the rostrocaudal axis (Supplementary fig. 2A). We next removed genes expressed by <10% of cells in their section (lowly expressed genes) and a log2-fold change value between sections below 0.2 (those that exhibited poor segment-specificity). This included group 13 genes, the expression of which triggers axial growth arrest, expressed at very low levels exclusively in the sacral samples, which included the coccyx[4] (Supplementary fig. 2B). With this approach, we derived a rostro-caudal *HOX* code comprising 18 genes that exhibited the most position-specific expression patterns across stationary cell types in the spine (Fig. 4A; Supplementary Data 3). Unexpectedly, this included the antisense gene *HOXB-AS3*, which exhibited strong sensitivity for positional coding of the cervical region amongst *HOX* genes ($p < 10^{-300}$). We validated our finding with two spatial sequencing technologies; Visium spatial transcriptomics (50µm resolution, whole transcriptome) and Caratana in-situ sequencing (single-cell resolution, 123-gene panel) (Fig. 4B–D; Supplementary fig. 2C; see methods).

Having identified this broadly applicable set of genes, we next investigated *HOX* expression in individual cell types (Supplementary Data 4). Osteochondral cells exhibited the broader *HOX* code, with the addition of *HOXB6*, which was specific to the cervical region. Similarly, in meningeal cells, *HOXC11* was specific to the sacrum, *HOXC5* the thorax and *HOXA5* the cervical region, with *HOXC4* demarcating cervical, rather than thoracic, tissue. *HOXB2* was not specific to cervical tissue. Tendon cells exhibited the greatest HOX specificity by region. In addition to the broader HOX code, *HOXC11* was specific to sacral cells, with *HOXB6*, *HOXA5*, *HOXB3 and HOXA3* specific to the cervical region. Muscle by contrast had a weaker HOX code, with *HOXD10, A11 & D11*

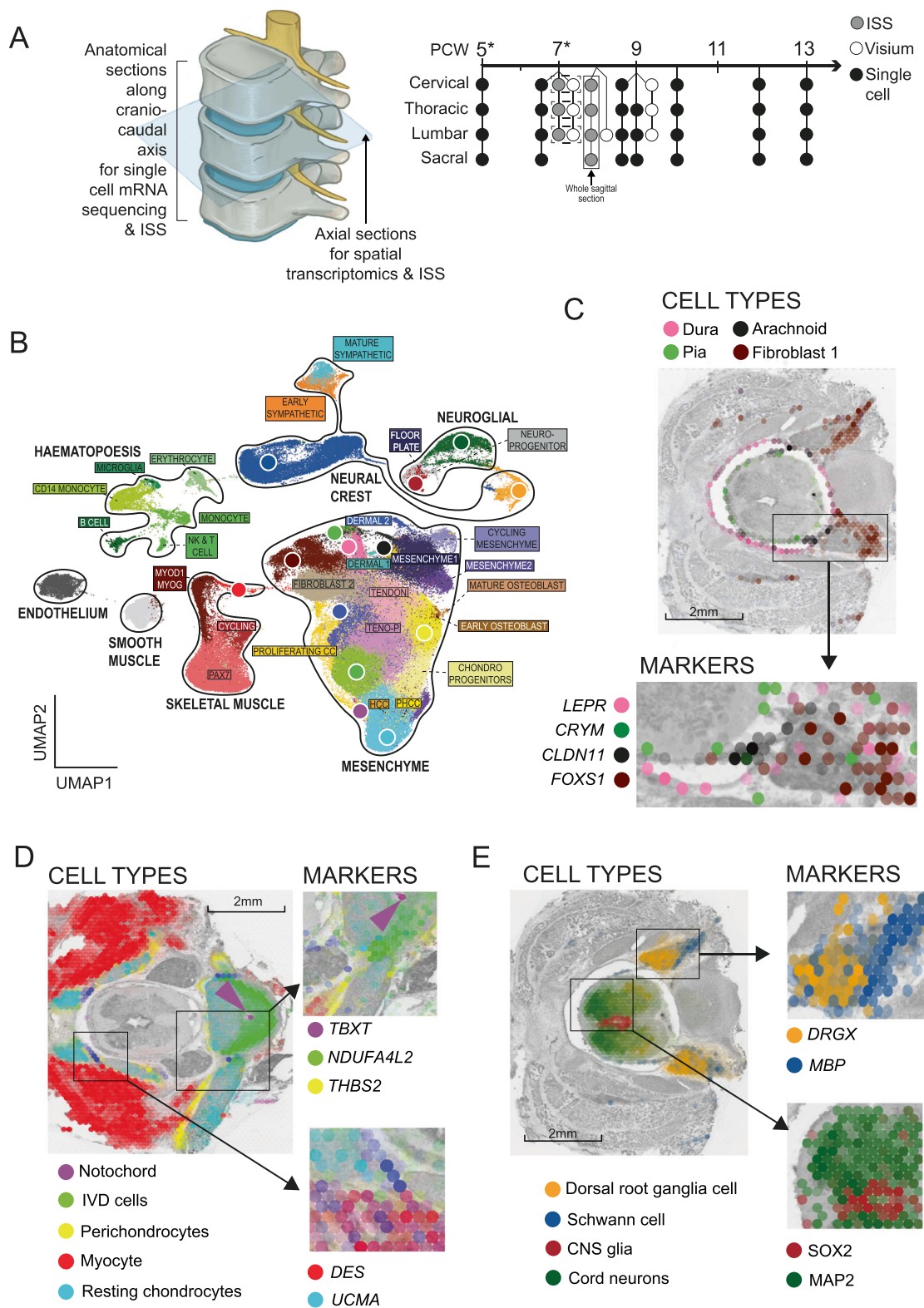

**Fig. 1 | An atlas of the developing fetal spine. A** Experimental overview. *Denotes samples that were not dissected into precise anatomical sections for scRNAseq. Dashed line denotes experiments performed on consecutive tissue sections; the solid box represents a whole sagittal section of the spine. PCW, Post-conception week; ISS, in-situ sequencing. **B** Uniform manifold approximation and projection of 175,000 fetal spinal cells. Coloured dots denote cell types demonstrated in **C–E** with the same colour. Teno-P, tenocyte progenitor; CC, chondrocyte; HCC, hypertrophic chondrocyte; PHCC, prehypertrophic chondrocyte. **C, D, E** Cell type mapping from single-cell to spatial transcriptomic samples at post-conception week 9 with heatmaps for marker genes. IVD, intervertebral disc cells; CNS, central nervous system. Source data are provided as a Source Data file.

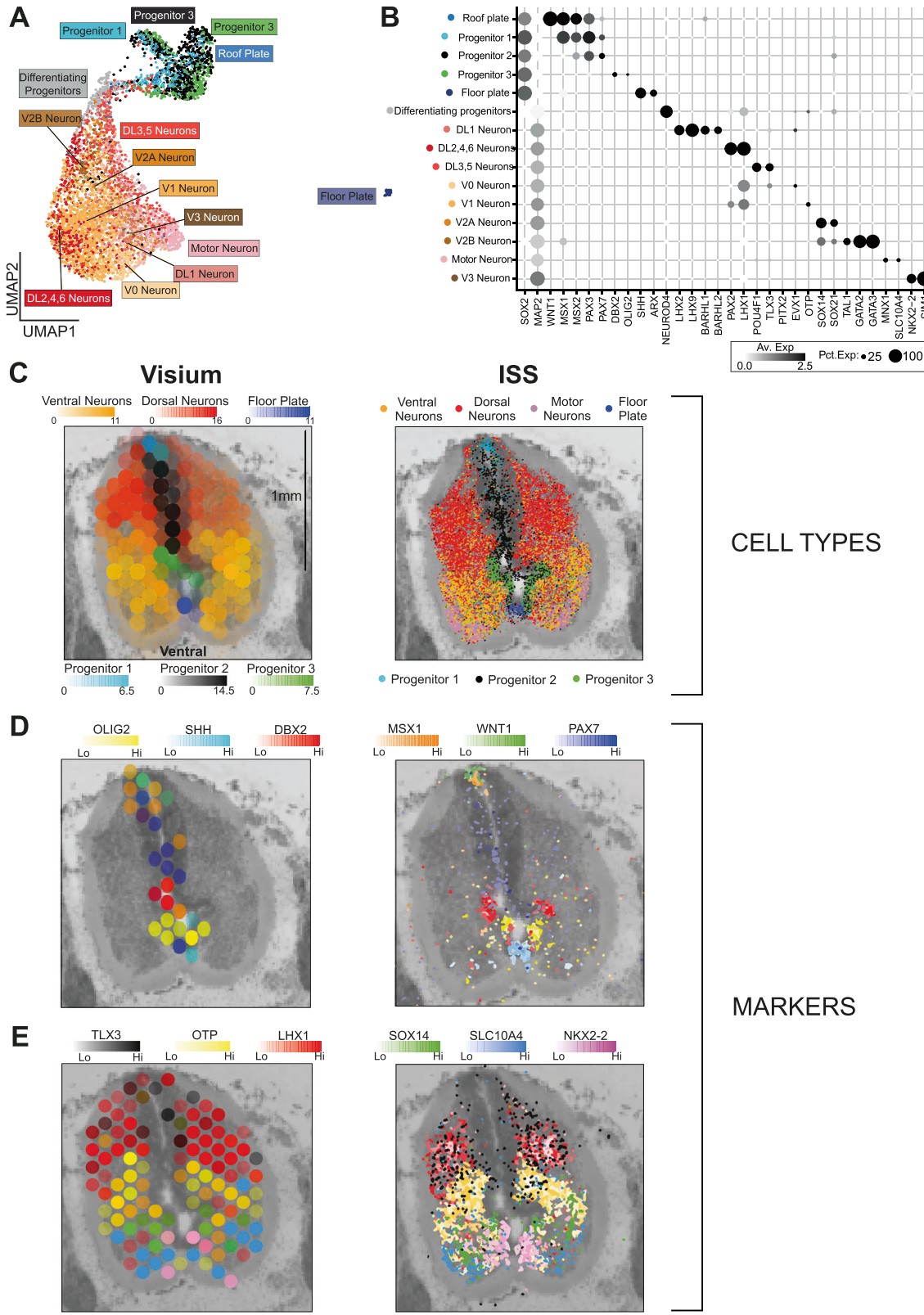

not reaching significance in the sacrum, *HOXA4 & B5* in the cervical region (which gained *HOXA3*), and only *HOXC4* and *HOXC6* maintained in the thoracic region. In fibrous tissue, *HOXD11* did not meet the logFC threshold in the sacrum, but instead *HOXC11* showed sacral specificity. *HOXD9* served as an additional lumbar region marker, and *HOXB3, A5 & C4* as cervical markers. With the exception of *HOXB-AS3 & HOXB8*, all other genes from the broader code remained. Finally, a few *HOX* genes

were significantly differentially expressed in vascular tissue within different regions(Fig. 4A; Supplementary Data 4). This perhaps reflects their partially migratory origins in early development[16]. Examining these cell type expression trends through the developmental period sampled, we noted a pattern of weakening of the *HOX* code in later developmental stages, with PCW13 consistently exhibiting fewer region-specific genes (Supplementary Data 4).

**Fig. 2 | Cellular landscape of the developing spinal cord. A** Uniform manifold approximation and projection of progenitors and neurons in the nascent cord. **B** Dotplot of mean expression values of marker genes amongst cells of the fetal cord. Dot size corresponds to the percentage of cells with non-zero expression. Pct. Exp, percentage of expressing cells; Av. Exp, average expression in cell type. **C** Left panel: Spatial heatmap showing cellular abundancy scores in the Post-conception week 7 spinal cord calculated using cell2location on spatial transcriptomics data. Right panel: inferred location of single cells in the PCW7 spinal cord using k-nearest neighbour prediction on in-situ sequencing data. ISS, in-situ sequencing. **D** Left panel: Spatial heatmaps of spatial transcriptomics data showing expression of marker genes for different neuroprogenitor populations from **A** & **B**; Right panel: gene expression of neuroprogenitor marker genes from in-situ sequencing data. **E** Left panel: Spatial heatmaps of spatial transcriptomics data showing expression of marker genes for different neuronal populations from **A** & **B**; Right panel: gene expression of neuronal marker genes from in-situ sequencing data. Source data are provided as a Source Data file.

This analysis allowed us to resolve *HOX* gene expression in cell types with poorly characterised (tendon cells) or controversial (spinal skeletal muscle[20]; spinal meningeal cells) *HOX* expression, charting their expression through development[21].

### Rostrocaudal *HOX* expression in neural-crest derivatives

Having established a core *HOX* code in stationary, mesenchyme-derived cell types, we assessed its utilisation in anatomically mobile and neural crest-derived (and therefore migratory) cell types. (Fig. 5A; Supplementary Fig. 3). As expected, anatomically mobile haemato-poetic cells did not follow the rostro-caudal *HOX* code. Examining cells expressing the neural crest markers *MSX1, FOXD3, SNAI1, SNAI2, SOX5, SOX8, SOX10 and ETS1*, we observed expression of the thoracic *HOX* code, in addition to the cervical gene *HOXB5*, across all anatomical locations, with no *HOX* genes differentially expressed in the thorax (Fig. 5A; Supplementary Data 3). In addition, these cells also expressed some *HOX* genes expressed by co-located stationary mesenchyme-derived cells. This included *HOXB2* in NCC-derived cells in the cervical region and *HOXC9, A9 & C10* to the lumbar region, together with *HOXD11* in the sacral region (Fig. 5A; Supplementary Data 3). However, NCC-derived cells also expressed *HOX* genes not differentially expressed by local mesenchyme-derived cells, including *HOXC4* in cervical cells, *HOXC8* in lumbar cells and *HOXA9 & C10* in sacral cells (Fig. 5A; Supplementary Data 3). In other words, these cells exhibited thoracic *HOX* expression regardless of their final destination (perhaps consistent with their truncal neural crest origin), whilst also expressing their own local *HOX* code, which partially overlapped with the expression pattern identified in stationary, mesenchyme-derived cells. This idiosyncratic *HOX* expression pattern seemed to remain consistent across the developmental stages sampled (Supplementary Fig. 3).

We validated this finding of thoracic *HOX* expression using both spatial transcriptomics (PCW9) and in-situ sequencing (PCW7) of axial sections at cervical, thoracic and lumbar levels (Fig. 5B). In the 50μm resolution spatial transcriptomics data, we found that voxels where the dominant cell type was NCC-derived expressed thoracic *HOX* genes, again with an expression of a local set of genes (Fig. 5B & D). For single-cell resolution in-situ sequencing data, we again showed the same expression pattern through analysis of *NTRK1*-expressing neurons within the dorsal root ganglia (DRG), which was first isolated anatomically using the WebAtlas browser's lasso tool, before subsetting for *NTRK1*-positive cells only (Fig. 5B–D; See Methods)[22]. In order to eliminate any potential batch / technical effect, we further validated this finding by performing RNA in-situ hybridisation on a whole spine sagittal section at PCW8, again showing expression of a thoracic *HOX* gene (*HOXB8*) in cervical and lumbar DRGs, alongside expression of a local *HOX* gene, though *HOXB4* did exhibit less restricted expression than *HOXC10* (Fig. 6A). Exploring this finding further within the in-situ sequencing data, we found that individual *NTRK1*-expressing NC-derived cells in the cervical and lumbar DRGs fell into four categories with regards their *HOX* expression pattern: 1) Truncal source code only; 2) local code only; 3) co-expressing or 4) no *HOX* expression (Fig. 6B–D). Thoracic neurons were categorised as source code or non-expressing (Fig. 6E).

The expression of a region-specific set of *HOX* genes in NCC-derived cells may commence before their migration begins, during their migration, or be adopted after they arrive in their new location. Our data may provide some clues in this regard by assessing, in individual cells, the correlation of the expression of the classical NCC marker *SOX10* versus local *HOX* code genes. If, for example, we found that more immature cells (i.e. those with higher expression of *SOX10*) expressed local *HOX* code genes less strongly, it may suggest that at least some of the local *HOX* code expression is acquired after neural crest settled in their new location and adopt their final identity. To pursue this analysis, we compared this ratio of *SOX10* versus local *HOX* code expression of the most immature and mature cells that we captured in sufficient numbers, i.e. lumbar cells at PCW9 and PCW12 (with the local *HOX* code *HOXC9, HOXA9, and HOXC10)*. Comparing PCW9 to PCW12 lumbar cells, we found a significant difference in the ratio for two *HOX* genes (*HOXA9* and *HOXC10*); more immature cells expressed relatively more *SOX10* than local *HOX* code (Fig. 6F). Although this observation would be consistent with a dynamic acquisition of the local *HOX* code upon arrival of neural crest cells, ultimately this question can only be answered through lineage tracing experiments. As these cannot be performed in human tissue, it would have to be pursued in model systems.

To further examine the expression of this thoracic *HOX* code through development, we performed scRNAseq on the lumbar spine from a PCW17 fetus. We found the expression of thoracic *HOX* genes, again with the expression of a set of lumbar genes that partially overlapped with that of other tissues, in neural crest-derived cells (Supplementary Fig. 4A-C). Finally, we examined *HOX* expression in a published adult lumbar (L4 & L5) DRG single nuclear sequencing dataset, and found consistent high expression of *HOXB8*, but not of other truncal *HOX* genes expressed by NCC in all regions of the fetal spine, in several clusters of *ELAVL3*+ neurons that differentially expressed the sensory neuron genes *NEFM* & *NEFH*[23,24] (Supplementary Fig. 4D & E). Other neural crest *HOX* genes were expressed at much lower levels in neuronal cells (Supplementary Fig. 4E).

### Validation in other developing organs

To extend and validate our observations about *HOX* expression in neural crest-derived cells, we examined neural crest derivatives in other fetal organs: the adrenal medulla, the hindlimb and the gut. In the adrenal medulla[25], where neural crest-derived Schwann Cell Precursors (SCP) generate sympathoblasts and chromaffin cells via bridge cells, we observed that these differentially expressed thoracic *HOX* genes, even after differentiation into mature cell types (Supplementary fig. 4F; Supplementary Data 5). By contrast, in the non-neural crest-derived adrenal cortex, *HOX* gene expression followed that of the adrenal portion of the adrenogonadal primordium, with a more 3' expression pattern[26] (Supplementary Fig. 4F). In the first-trimester fetal hindlimb, mesenchymal tissues (muscle, cartilage and tendon) of the stylopod and zeugopod expressed an anatomically appropriate code of groups 9-11, whereas SCPs expressed thoracic/truncal *HOX* code in addition to expressing this local code[27] (Supplementary Fig. 4G; Supplementary Data 5). In the fetal gut, cells of the neural-crest-derived enteric nervous system exhibited the 3' *HOX* code of their origin – that of the vagal neural crest, regardless of anatomical location[28] (Fig. 6G). By contrast, mesenchymal cells expressed a regionally-appropriate *HOX* code, with colonic cells differentially expressing lumbosacral genes when compared to cells

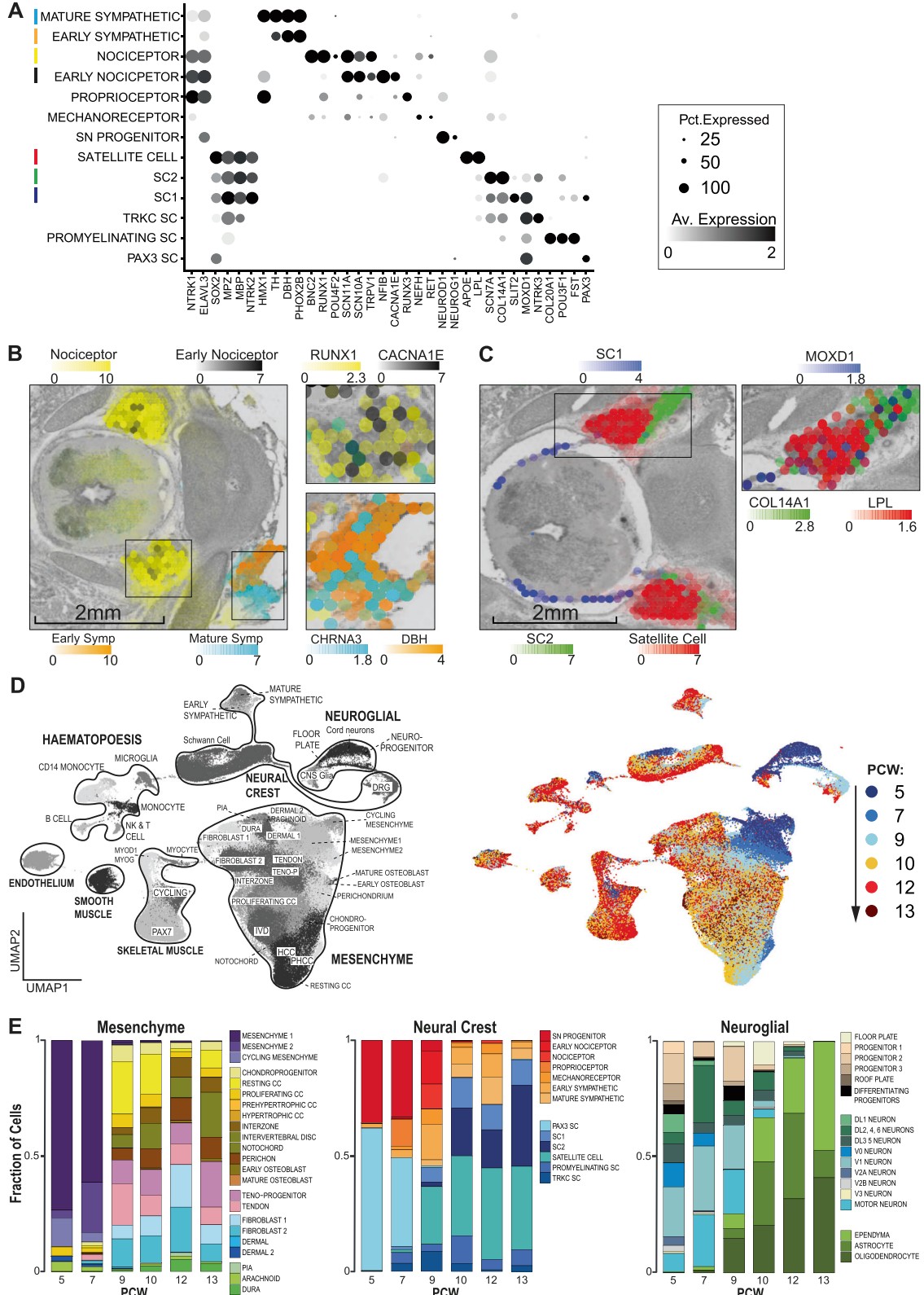

of the ileum, which expressed a strong 3' signal[28,29] (Fig. 6G; Supplementary Data 5). Taken together, these findings delineated a tissue overarching principle of thoracic *HOX* gene expression in neural crest derivatives; a significant portion of these cells seem to express *HOX* genes that correspond to their anatomical region of origin in the neural crest.

## Axial *HOX* gene expression in the developing spinal cord

Next, we searched for patterning genes along other body axes – left versus right and dorsoventrally. We first compared in each Visium section transcription of the same cell type in different anatomical side (eg, skeletal muscle-dominated voxels on left versus skeletal muscle voxels on right), and asked whether there were cell type independent

**Fig. 3 | Cellular landscape of the peripheral nervous system. A** Dotplot of mean log-normalised expression values of marker genes amongst cells of the fetal peripheral nervous system. Point size shows the percentage of cells with non-zero expression. Coloured bars correspond to cell types represented in **B**. SC, Schwann cell; SN, sensory neuron; Pct. Exp, percentage of expressing cells; Av. Exp, average expression in cell type. **B** Spatial heatmaps showing cellular abundancy scores (left panel) in Post-conception week 9 peripheral nervous system calculated using cell2location and marker gene expression for those cell types (right panel). Boxes on the left panel denote regions represented in two panels on the right. Symp, sympathetic neurons. **C** Spatial heatmaps showing cellular abundancy scores (left panel) in PCW9 peripheral nervous system calculated using cell2location and marker gene expression for those cell types (right panel). Box on the left panel

denote regions represented in the panel on the right. SC, Schwann cell. **D** Uniform manifold approximation and projection (UMAP) plot visualising single-cell RNA sequencing data cell types (left panel) and contribution of each post-conception week stage to the manifold (right panel). PCW, post conception week;Teno-P, tenocyte progenitor; IVD, intervertebral disc; HCC, hypertrophic chondrocyte; PHC, prehypertrophic chondrocyte; DRG, dorsal root ganglia; CC, chondrocyte; CNS, central nervous system. Colours in the bar chart represent cell types in the adjacent colour key. **E** Stacked bar charts showing the fraction of each cell type in the mesenchymal (left panel), neural crest (centre panel) and neuroglial (right panel) compartments at each sample stage. CC, chondrocyte; SC, schwann cell; SN, sensory neuron; PCW; post-conception week. Source data are provided as a Source Data file.

transcripts that distinguished these planes. Whilst these analyses did not yield candidate genes that pervade all cell types, they did highlight other complex patterns of spatially variable gene expression in the spinal cord[30]. We found that the expression of *HOXB* genes in the human embryonic cord did not exhibit collinearity, with members 3-9 of the cluster to the dorsal cord regardless of anatomical level (Fig. 7A & B; Supplementary Fig. 5A). Intriguingly, this included the antisense transcript *HOXB-AS3*, which was expressed in the medial aspect of the dorsal horn and the periaqueductal region (Fig. 7A; Supplementary Fig. 5A). Whilst group B genes were all dorsally expressed, their precise expression patterns varied within this domain (Fig. 7A). These findings build on patterns of *HOXB* expression identified in earlier developmental stages in animals, whereby *HOXB* genes are initially ubiquitously expressed throughout the neural tube, then restricted to the intermediate zone, which traverses the dorsoventral axis of the neural tube[31,32]. Analysis of a single publicly available PCW5 spatial transcriptomics axial section suggested that at this earlier stage *HOXB8* expression did not exhibit such clear dorsal restriction (Supplementary fig. 5B)[33]. At PCW12, *HOXB8* expression was very similar to PCW9 with clear restriction to the dorsal cord (Supplementary fig. 5B).

The identity taken up by ventral progenitors in the cord is thought to be highly dependent on *HOX* expression. This was recently highlighted in vitro by Mouilleau et al. who showed that stalling FGF-induced temporal activation of *HOX* genes in human pluripotent stem cell (hPSC)-derived progenitors resulted in cervical motor neuron types, whereas accelerating *HOX* progression produced lumbar neuronal subtypes[34]. We therefore aimed to explore these trends in vivo by examining *HOX* expression in the fetal ventral spinal cord along the rostro-caudal axis. In contrast to the dorsal horns, in the ventral spinal cord *HOX* expression varied with rostrocaudal position, highlighting the role of these genes in establishing anatomically appropriate motor neuron (MN) pools (Fig. 7B)[35,36]. *HOXC5 & 6* were expressed in the ventral cervical cord, with the latter extending somewhat into the thoracic cord (Fig. 7B). *HOXC8 & 9* were expressed in the ventral thoracic cord, with the former exhibiting some ventral cervical expression (Fig. 7B). This overlap of *HOXC6 & 8* is most likely due to their combined role in patterning digit-innervating neurons as the caudal limit of the cervical cord[36,37]. *HOXC10 & HOXD10* were exclusive to the ventral lumbar cord (Fig. 7B).

In order to explore these patterns further, we performed in-situ sequencing on a full-length midline sagittal section through the spine at PCW 8 (Fig. 8A). This again confirmed restriction of *HOXB8* to the dorsal cord, with minimal ventral expression (Fig. 8A). *HOXC6* exhibited diffuse expression throughout the ventral cord, extending more rostrally than other genes (Fig. 8A). Together with *HOXC9*, it was also expressed throughout the dorsal cord, at comparable levels (Fig. 8A). As in the ST data, *HOXC8* exhibited a more rostral superior limit than *HOXC9*, in keeping with its dual role in patterning the caudal half of the lateral motor columns (LMC), including *ETV4*[+] MNs, through co-expression with *HOXC6*[38], which was detected only at the inferior limit of the cervical cord (Fig. 8A). Interestingly, *HOXC6* was expressed in the same territory as *HOXC9, and HOXC8/9* expression extended into

the territory of *HOXC10/D10* expression (Fig. 7B, Fig. 8A). Whilst spatial transcriptomics data confirmed that *HOXC6* expression dominates the ventral cervical cord, and group 10 genes dominate the ventral lumbar cord, both datasets revealed the surprising maintenance of lower level expression of other motor-column *HOX* genes in these regions (Figs. 7B and 8A). In the axial ST sections, at PCW7 & 9, multiple voxels covering the ventral cervical and thoracic cord coexpressed *HOXC6* and *HOXC9*, and several voxels covering the ventral lumbar cord coexpressed *HOXC9* and *HOXC10/D10* (Fig. 7B, white asterisks). Further analysing this at single-cell resolution using ISS at PCW7 (from sections 12 μm adjacent to the Visium samples), we confirmed that some ventral cells at each level did indeed express, in combination, motor column-defining *HOX* genes previously reported as mutually exclusive (Fig. 7C). The same was true for the sagittal sections at PCW8 (Fig. 8B).

Finally, we also examined expression of these genes by region in the single cell data (almost all of which originated from week 5 & 7 samples), again finding *HOXB* genes were exclusive to dorsal neurons in the developing cord with a high percentage of cells expressing them (Fig. 8C). The ventral *HOX* genes showed mixed expression (Fig. 8C). *HOXC5, HOXC8, HOXC10 & HOXD10* were exclusive to ventral neurons, expressed by small percentages of the total cells (presumably reflecting their restriction along the rostro-caudal axis to particular regions), but *HOXC6* and *HOXC9* were more dorsally expressed (Fig. 8C). Together, These findings suggest that the mutual cross-repression of these groups of motor-defining *HOX* genes in the ventral cord is incomplete during the period studied (PCW7-9).

In a final analysis we searched for other rostrocaudal patterning genes that follow the expression of the core *HOX* code, that is, genes with rostrocaudal expression gradients across all stationary cell types that correlated with the core *HOX* code. Although our analyses did not reveal such genes, they enabled us to identify genes that varied with rostrocaudal position in some cell types. This included *LINC01497*, enriched in cervical perichondrial cells, which is associated with both short stature and decreased bone mineralisation, and potential negative regulators of osteoblastgenesis enriched in lumbar cells, such as the sulphating enzyme *SULT1E1*, which plays a key role in oestrogen metabolism, and the cell adhesion molecule *NEGR1*[39–41] (Supplementary fig. 7D & Supplementary Data 6). *CHST9*, which catalyses sulfation of proteoglycans, was enriched in cervical resting chondrocytes, along with *HHIP*, a regulator of hedgehog signalling[42] (Supplementary fig. 7D). Interestingly, lumbar chondrocytes differentially expressed *TMSB4X*, a molecule implicated in load-adaptation (Supplementary fig. 7B)[43]. Examining the *HOX* regulons using the DoRothEA R package, we found that none of these genes had direct evidence of being HOX targets[44].

## Discussion

Here, we built a developmental cell atlas of the human spine based on high-resolution, quantitative transcriptional readouts. We were thus able to study the expression of *HOX* genes in human cell types along the rostrocaudal axis, as well as in the axial plane, which confirms and

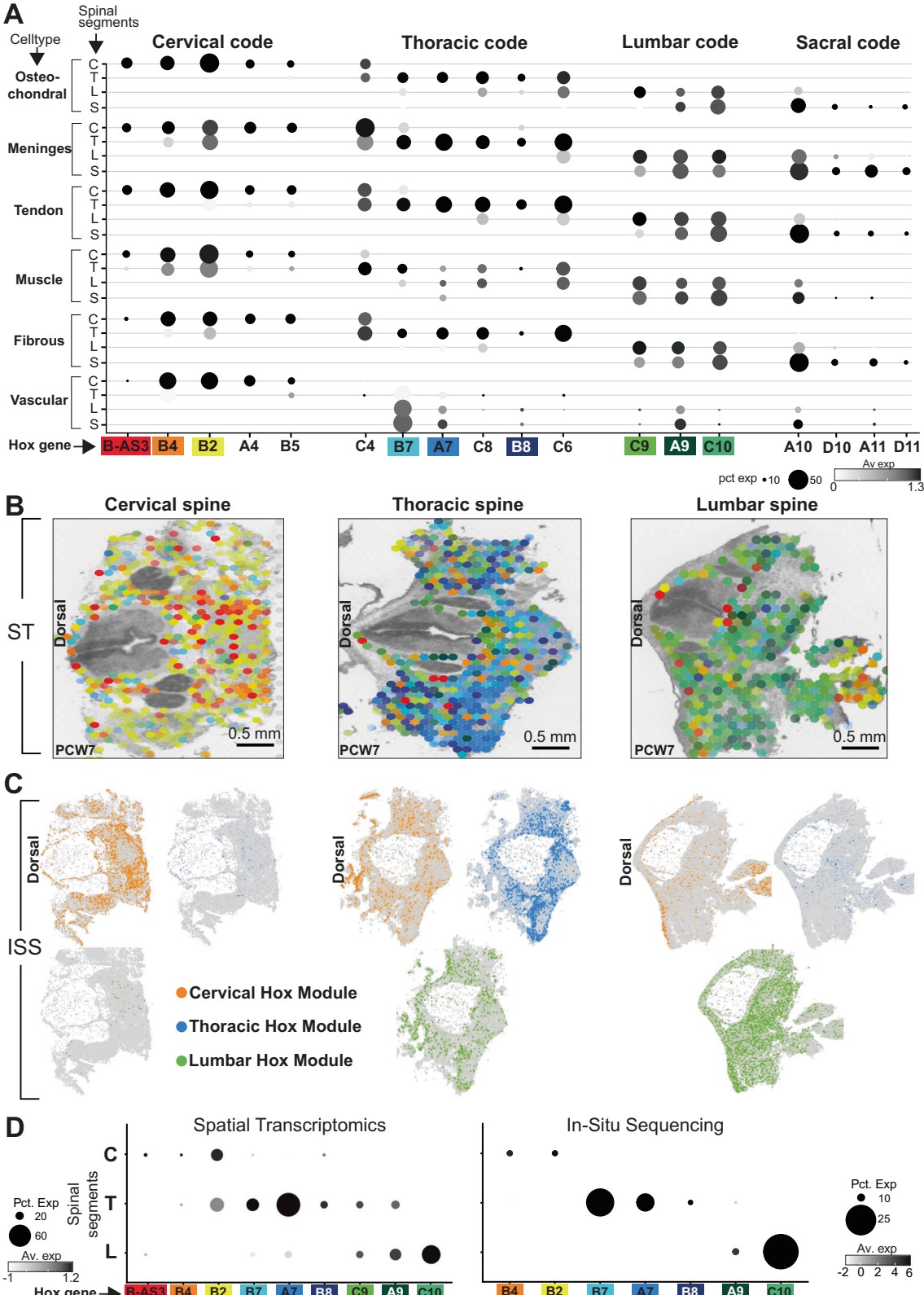

**Fig. 4 | The rostro-caudal HOX code. A** Dotplots quantifying log-normalised *HOX* gene expression across spinal segments in stationary cell types. Dot size corresponds to the fraction of cells with non-zero expression. Coloured boxes highlight genes represented in **B**. C, cervical; T, thoracic; L, lumbar; S, sacral; Pct. Exp, percentage of expressing cells; Av. Exp, average expression in cell type. **B** Spatial heatmaps of *HOX* gene expression across spinal segments in stationary cell types at PCW7 from spatial transcriptomics data. Dot colour in the plot corresponds to the genes coloured in (A). PCW, post-conception week. **C** Expression of *HOX* gene modules from in-situ sequencing across spinal segments. ISS, in-situ sequencing. Dot colour corresponds to expression of the gene module highlighted in the legend (orange- cervical; blue- thoracic; green- lumbar). **D** Dotplots quantifying log-normalised *HOX* gene expression levels for spatial transcriptomics and in-situ sequencing data. Left panel: spatial transcriptomics data. Right panel: in-situ sequencing data. Dot size corresponds to the fraction of voxels/cells with non-zero expression. C, cervical; T, thoracic; L, lumbar. Coloured boxes highlight genes represented in **B**. Source data are provided as a Source Data file.

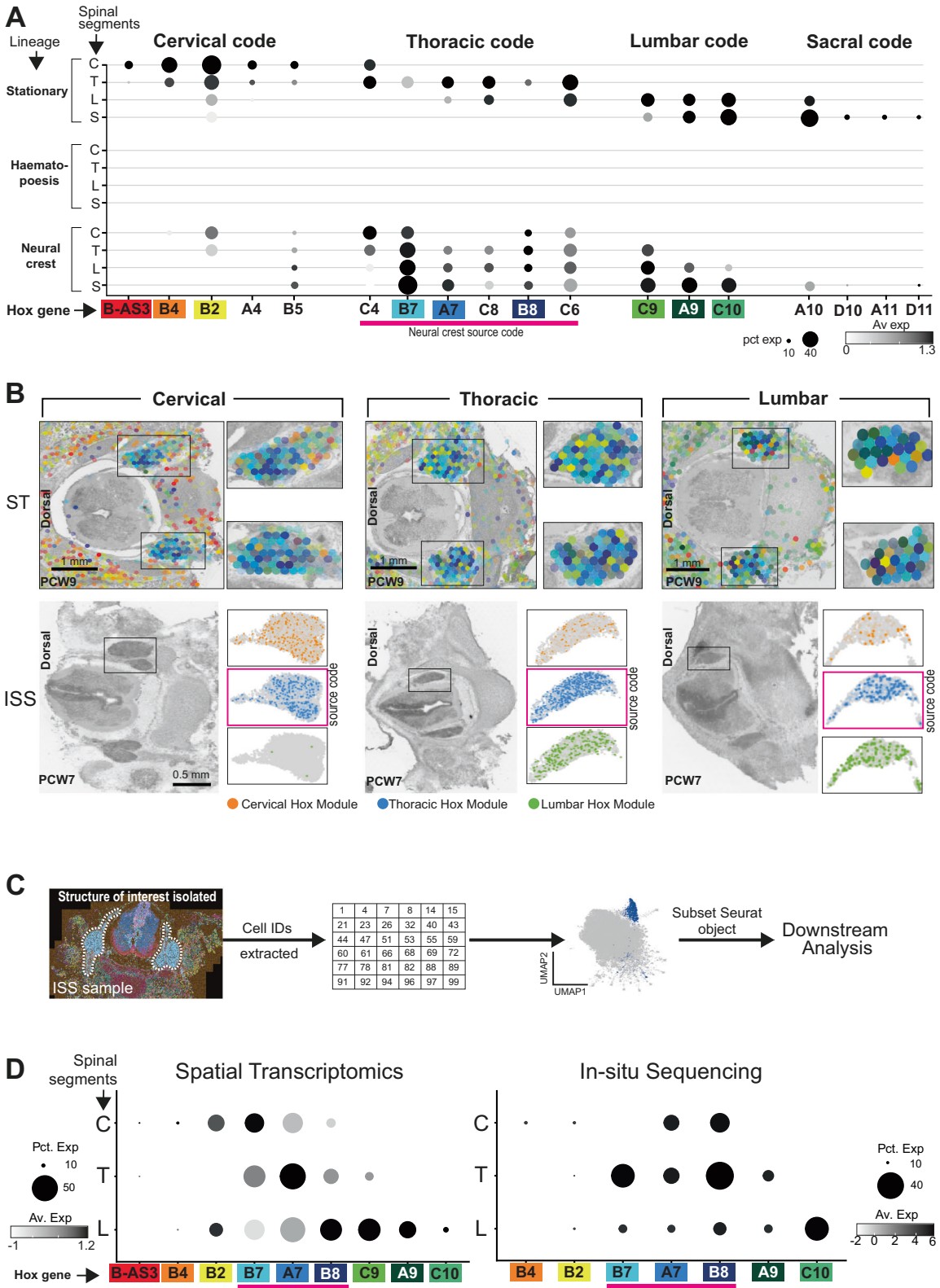

extends, in humans, the canon of *HOX* biology derived over decades from model organisms. In addition to serving as positional codes, *HOX* genes perform additional lineage and cell-type specific functions in development[11,12,45]. As a consequence, some *HOX* genes lack specificity to serve as positional anchors for analyses. Our work distilled from orthogonal methods and multiple independent specimens a core *HOX* code in mesenchyme-derived tissues in the human spine between the

9th and 13th weeks postconception. Interestingly, we found that during this developmental window, NCC-derived cells seemed to express a thoracic *HOX* code, perhaps in keeping with their truncal origin within the neural crest, whilst also expressing a set of region-specific *HOX* genes at their end destination. Whilst there was some overlap between these region-specific genes and the anatomical code derived from mesenchyme-derived cells, differences were also evident. This

**Fig. 5 | HOX gene expression in neural crest-derived cells. A** Dotplots quantifying log-normalised *HOX* gene expression across spinal segments in stationary, hae-matopoietic and neural crest cell types. Dot size corresponds to the fraction of cells with non-zero expression. Coloured boxes highlight genes represented in **B**. C, cervical; T, thoracic; L, lumbar; S, sacral; Pct. Exp, percentage of expressing cells; Av. Exp, average expression in cell type. **B** Top panel: Spatial heatmaps of *HOX* gene expression in neural crest cell types at PCW9 from spatial transcriptomics data. Dot colour represents genes coloured in **A**. Bottom panel: Expression of *HOX* gene modules from in-situ sequencing across spinal segments. Dot colour corresponds

to expression of the gene module highlighted in the legend (orange- cervical; blue-thoracic; green- lumbar). PCW = Post-conception week. ISS = in-situ sequencing. **C** Schematic illustrating workflow for analysis of in-situ sequencing data at single cell resolution. ISS, in situ-sequencing. **D** Dotplots quantifying log-normalised expression levels in **B**.Left panel: spatial transcriptomics data. Right panel: in-situ sequencing data. Dot size corresponds to the fraction of voxels/cells with non-zero expression. C, cervical; T, thoracic; L, lumbar. Source data are provided as a Source Data file.

suggests that NCC-derived cells utilise a different *HOX* code to mesenchyme for positional information. This idiosyncratic expression pattern is in keeping with other human tissues, such as the mesenchyme of the limb bud (which has its own *HOX* expression patterns) and the ventral and dorsal cord as discussed here[46]. We showed this pattern of *HOX* expression corresponding to neural crest origin held true in the fetal limb and adrenal gland (containing truncal neural crest-derived cells), and the fetal gut (containing vagal neural crest-derived cells). Together, these findings shed further light on the utilisation of this enigmatic set of genes by different tissues during development.

The functional consequences of this neural crest *HOX* 'source code' remain unclear. Using in-situ sequencing, we demonstrated that *NTRK1*-expressing DRG neurons fall into several categories in terms of *HOX* expression: non-expressing, local code expressing, source code expressing and co-expressing. This may reflect functional subsets of neurons, with particular *HOX* combinations dictating subsequent biological processes, as is the case for the proprioceptive subset of sensory neurons in mice, in which *HOX* expression dictates sensory-motor synaptic matching[47]. It may be that similar *HOX*-driven developmental processes also take place in the non-neuronal NCC-derived tissue studied here, such as the sympathoblasts of the adrenal gland.

Our study also investigated the expression of *HOX* genes along the dorsoventral axis of the spinal cord. Previous work with model organisms has uncovered the evolving role of the *HOXB* genes in neurogenesis. In the neural tube of the mouse and chick, *HoxB* genes are expressed before other *Hox* clusters, initially in a non-specific manner throughout the tube, before becoming somewhat restricted to the intermediate zone; an area bridging the ventricular zone (containing progenitors) and the mantle zone (containing post-mitotic cells)[31,32]. Our study continued the study of this evolving expression pattern, showing with both single cell and spatial transcriptomics that at later time points (in the human) *HOXB* genes show restriction to the dorsal cord, and continue to lack any rostrocaudal collinearity. Perhaps after regulating neuronal delamination via *LZTS1* at the intermediate zone in the neural tube, *HOXB* genes take on a further role in specifying sensory / dorsal identity in the cord. Further experiments in model systems examining the effect of *HOXB* inactivation at different developmental stages would shed light on their role at the stages studied here.

A further interesting finding in our study was the co-expression of certain motor-pool determining *HOX* genes in the ventral cord, which had previously been reported as mutually exclusive in animal models. As discussed above regarding *HOXB* gene expression, this may again be a reflection of evolving patterns of *HOX* expression during development; our study window being comparatively late compared to the animal models. However it remains possible that this is a species-specific feature. A previous in vitro study by Mouilleau et al. used the timing of retinoic acid application to culture to recapitulate the temporal nature of motor neuron development; cervical neurons develop earlier than lumbar neurons[34]. Adding retinoic acid at day 3 onwards resulted in strong *HOXC6* expression, whereas addition at day 6 onwards resulted in a clear shift towards *HOXC9*, recapitulating expression along the rostrocaudal axis[34]. However, addition at day 5 did seem to result in some overlap in the expression of these two

genes, though co-localisation analysis of these two supposedly exclusive genes was not performed. Further studies of their expression dynamics using similar methods would be enlightening. Finally, whilst RNA assays provide high resolution and sensitivity, they do not necessarily reflect the presence of protein; it may be that at the protein level *HOX* distribution in the human fetal cord is identical to that of animal models.

When examining *HOX* gene expression along the rostrocaudal axis, we utilised data generated from fetuses dissected into anatomical sections. This analysis was limited by the relatively narrow sample window (PCW9-13), as the earliest samples (PCW5 & 7) could not be reliably dissected into precise anatomical sections. This meant that, whilst robust across these relatively late developmental stages, it remains unclear how *HOX* gene expression across cell types changes early in human development. Model organism work has shown, in detail, the dynamic nature of *HOX* expression in the neural tube and early cord, and investigation of these periods using the techniques we employed here would be enlightening, if logistically challenging due to the process of fetal tissue collection from termination of pregnancy[31,32]. In the stationary cell types captured by our experiments, it would be similarly interesting to examine earlier developmental stages for any transient *HOX* expression patterns, such as the waves of expression in the nascent limb bud[27].

A further limitation of our study is the breadth of cell capture per fetal spinal section in the scRNAseq data. In particular, neuroglial cells were poorly represented across ages and sections, preventing in-depth analyses of these cell types. Further studies with similar methodology but enriched for neuroglial cells would allow further investigation of *HOX* patterns in this tissue. Similarly, when investigating the dynamics of *HOX* expression in NCC-derived cells, we were limited to two samples (lumbar PCW9 & 12) as no other two samples from different fetuses at the same anatomical level captured sufficient NCC-derived cells to allow adequately powered analysis. Furthermore, PCW9 is relatively advanced in terms of neural crest specification; studying earlier time points would shed further light on *HOX* expression dynamics. Conversely, whilst spatial transcriptomics experiments capture RNA from a whole tissue section, the resolution of the technique in its current form (50μm, with dead space between voxels) is limiting. In our study, this prevented detailed analysis of PCW7 *HOX* expression at the DRG, as very few voxels covered the DRG, and many that did had significant overlap with non-neuronal tissue. This was particularly the case in the lumbar sample, which had the smallest DRGs. We were thus dependent on the much larger PCW9 samples for single transcriptomics-based validation of the *HOX* source code and unable to directly compare spatial transcriptomics and in-situ-sequencing results at PCW7. Similarly, the much smaller size of PCW7 samples meant individual voxels were more likely to cover multiple tissue types, meaning PCW9 samples were preferable for mapping sequenced single cells to their predicted location on the ST slides.

High-resolution transcriptional assays of human fetal cells represent a powerful tool for assessing differentiation states. However, transcription is by definition plastic and therefore does not usually generate fixed developmental barcodes (e.g. somatic mutations),

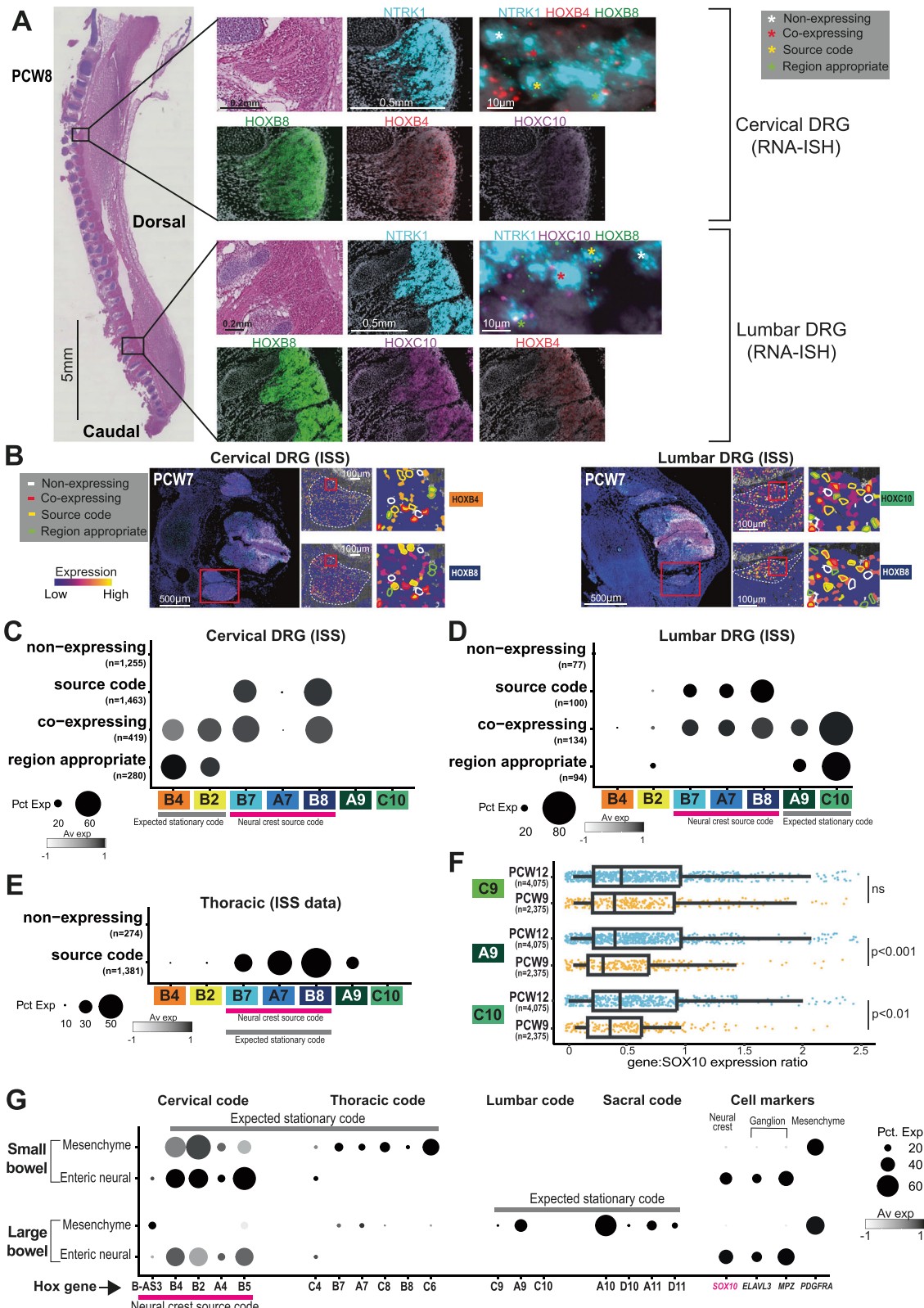

which lineage tracing relies upon. Our finding of *HOX* gene expression in truncal and vagal neural crest-derived cells that corresponds to their point of origin may lend itself to retrospective lineage tracing in development from mRNA assays. As the developmental Human Cell Atlases will be extended in coming years, identification of idiosyncratic gene expression patterns in cells of particular developmental origins may provide a powerful tool to obtain a fuller picture of developmental patterning in humans, not confined to the stages and organ systems we studied here.

## Methods

### Ethics statement

Human embryonic tissue was collected from elective termination of pregnancy procedures at Addenbrookes Hospital, Cambridge, UK

**Fig. 6 | Categorising dorsal root ganglion (DRG) neurons by *HOX* gene expression. A** RNA-In Situ Hybridisation of the PCW8 whole spine in the para-sagittal plane for *HOXB8, HOXB4, HOXC10 & NTRK1*. The inset high magnification boxes represent the region in the corresponding box on the lower magnification image. Rectangular boxes show higher magnification images from each region with examples of *NTRK1*+ve cells that express either none of the *HOX* genes stained for (white asterisk), both local *HOX* gene and *HOXB8* (red asterisk), *HOXB8* only (yellow asterisk) or the local *HOX* gene only (green asterisk).DRG, dorsal root ganglion; PCW, post-conception week. **B** In-situ sequencing of the PCW7 spine, including dorsal root ganglion in the axial plane. Red rectangular boxes indicate higher magnification image areas from each region with highlighted examples of *NTRK1*+ve cells that express either none of the *HOX* genes (white outlines), both local *HOX* gene and *HOXB8* (red outline), *HOXB8* only (yellow outline) or the local *HOX* gene only (green outline).DRG, dorsal root ganglion; ISS, in-situ sequencing; PCW, post-conception week. **C** Dotplot of mean log-normalised expression values of *HOX* genes amongst cervical DRG neurons in in-situ sequencing data, categorised by expression pattern. Point size shows the percentage of cells with non-zero expression. ISS, in-situ sequencing; Pct Exp, Percentage of cells expressing; Av exp, Average expression in group. **D** Dotplot of mean log-normalised expression

values of *HOX* genes amongst lumbar DRG neurons, categorised by expression pattern. Point size shows the percentage of cells with non-zero expression. ISS, in-situ sequencing; Pct Exp, Percentage of cells expressing; Av exp, Average expression in group. **E** Dotplot of mean log-normalised expression values of *HOX* genes amongst thoracic DRG neurons, categorised by expression pattern. Point size shows the percentage of cells with non-zero expression. ISS, in-situ sequencing; Pct Exp, Percentage of cells expressing; Av exp, Average expression in group. **F** Box plot quantifying expression ratios of *HOXC9, A9 & C10* vs *SOX10* in neural crest-derived cells at PCW9 and 12 in the lumbar regions. The box contains the 25th to 75th percentiles of the data, with the central line denoting the median value. The upper whisker extends from the median to the largest value that is no further than the 1.5* inter-quartile range (IQR). The lower whisker extends from the median to the smallest value, at most 1.5* IQR. Wilcoxon rank-sum test, two-sided, with adjustment for multiple comparisons. PCW, post-conception week. Orange dots represent PCW9 cells, blue dots represent PCW12 cells. **G** Dotplot quantifying log-normalised *HOX* gene and classical marker gene expression in stationary and neural crest cell types within the gastrointestinal system. Dot size corresponds to the fraction of cells with non-zero expression. Pct Exp, Percentage of cells expressing; Av exp, Average expression in group. Source data are provided as a Source Data file.

under full ethical approval from the East of England–Cambridge Central Research Ethics Committee (REC-96/085) and from the joint MRC/Wellcome Trust Human Developmental Biology Resource (htttp://hdbr.org; grant number MR/R006237/1) at the Institute of Child Health, London (Registration number 200532; REC 18/LO/ 0822) under full ethical approval from the London-Fulham Research Ethics Committee. Informed, written consent was obtained from patients after the decision was made in the clinic to terminate pregnancy, in advance of the termination of pregnancy procedure. Experiments followed the 2021 International Society for Stem Cell Research (ISSCR) guidelines in working on human embryos. No developmental abnormalities were visible or known in any of the embryos collected. No participant compensation occurred. In both cases, fetal age (postconception weeks, PCW) was estimated using the independent measurement of the crown rump length (CRL), using the formula PCW (days) = 0.9022 × CRL (mm) + 27.372. Genotyping for embryo sex was not performed.

### Tissue dissection and dissociation

Fetal spines were dissected under a microscope using sterile microsurgical instruments. Where anatomy was reliably preserved (PCW9 onwards), the spine was dissected into separate anatomical sections (cervical, thoracic, lumbar and sacral) by an orthopaedic surgeon (JEGL) prior to digestion. The cervico-thoracic junction was identified by the junction of the 1st rib to the body of the first thoracic vertebra. A cut was then made at the C7/T1 disc to isolate the cervical section. Vertebrae were then counted rostrocaudally to identify the first lumbar vertebra, sense-checked by the presence of the 12th rib joining the 12th thoracic vertebral body immediately superior to it. A cut was then made at the T12/L5 disc to isolate the lumbar and thoracic sections. Finally, a cut was made at the L5/S1 disc to isolate the sacro-coccygeal section. Each sample was digested in a 5 µg/ml Liberase TH working solution prepared from Liberase TH powder (Sigma 5401135001) and 1X Phosphaste Buffered Saline (PBS) on a shaking platform (750 rpm) at 37 °C for 30 min. The tissue was gently agitated using a P1000 pipette after 15 min. 5 ml 2% Fetal bovine serum (FBS) in PBS was then added to stop the dissociation, prior to second stage digestion with 0.25% trypsin solution for a further 30 min at 37 °C, with pipette-agitation every 5 min. Cells were then spun down at 750 g at 4 °C for 5 min and resuspended 50-200ul of 2% FBS in PBS. Fetal cells were loaded for scRNAseq directly following sample processing.

### Visium spatial transcriptomics experiments

Embryonic spine samples from the cervical, thoracic and lumbar regions at PCW7-9 were embedded in OCT within cryo wells and flash-

frozen using an isopentane & dry ice slurry. Ten-micron thick cryo-sections were then cut in the axial plane and transferred onto Visium slides prior to haematoxylin and eosin staining and imaged at 20X magnification on a Hamamatsu Nanozoomer 2.0 HT Brightfield. These slides were then further processed according to the 10X Genomics Visium protocol, using a permeabilisation time of 12 min for all samples. Images were exported as tiled tiffs for analysis. Dual-indexed libraries were prepared as in the 10X Genomics protocol, pooled at 2.25 nM and sequenced 4 capture areas per Illumina Novaseq SP flow cell with read lengths 28 bp R1, 10 bp i7 index, 10 bp i5 index, 90 bp R2. For PCW7 samples, in order to reduce technical batch effects, two anatomical regions were included on each capture areas (cervical & lumbar; cervical & thoracic; thoracic & lumbar) totalling 3 capture areas. For PCW9, a single capture area was used per anatomical region, with two thoracic sections taken (upper and lower), totalling 4 capture regions. A single lumbar PCW8 section was sequenced on the final capture area.

### Cartana in-situ sequencing experiments

In situ sequencing was performed using the 10X Genomics CARTANA HS Library Preparation Kit (1110-02, following protocol D025) and In Situ Sequencing Kit (3110-02, following protocol D100). Embryonic spine samples from the cervical, thoracic and lumbar regions at PCW7 were embedded in OCT and axially sectioned as described above. Sections immediately adjacent to those used for the Visium assay

ISS was performed using the 10X Genomics CARTANA HS Library Preparation Kit (1110-02, following protocol D025) and In Situ Sequencing Kit (3110-02, following protocol D100), which comprise a commercialised version of HybISS[48].

Briefly: cryosections of developing spine samples were fixed in 3.7% formaldehyde (Merck 252549) in PBS for 30 min and washed twice in PBS for 1 min each prior to permeabilization: sections were briefly digested with 0.5 mg/ml pepsin (Merck P7012) in 0.1 M HCl (Fisher 10325710) at 37 °C for 1 min, then washed twice again in PBS, all at room temperature. Following dehydration in 70% and 100% ethanol for 2 min each, an appropriate SecureSeal hybridisation chamber (either 9 mm diameter, 50 µl volume; 13 mm diameter, 100 µl; or 21.5 × 34.5 mm, 400 µl; all Grace Bio-Labs) was adhered to each slide and used to hold subsequent reaction mixtures. Following rehydration in buffer WB3, probe hybridisation in buffer RM1 was conducted for 16 h at 37 °C. The 123-plex probe panel included 5 padlock probes per gene, the sequences of which are proprietary (10X Genomics CARTANA). The section was washed with PBS-T (PBS with 0.05% Tween-20) twice, then with buffer WB4 for 30 min at 37 °C, and thrice again with

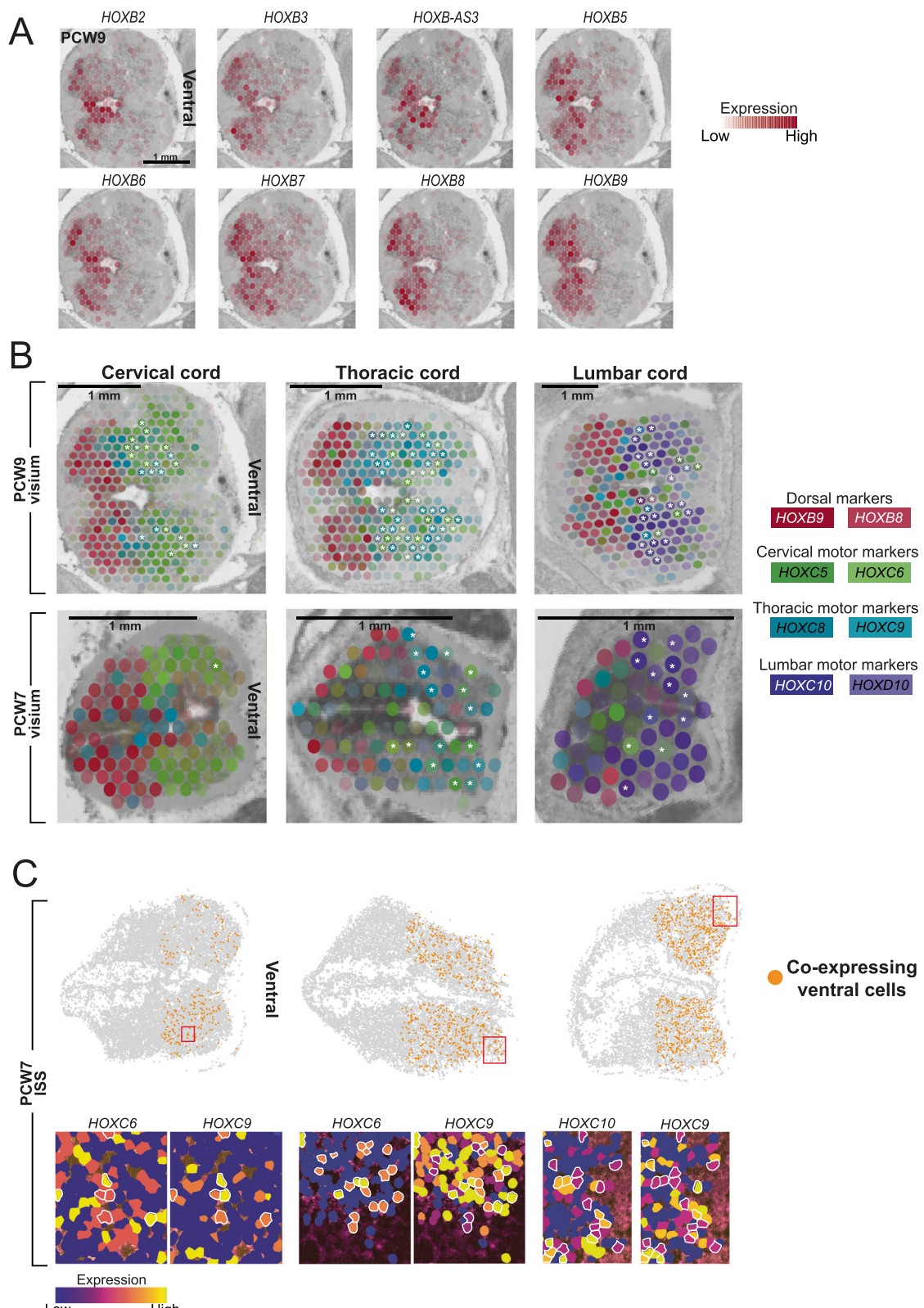

PBS-T. Probe ligation in RM2 was conducted for 2 h at 37 °C and the section washed thrice with PBS-T, then rolling circle amplification in RM3 was conducted for 18 h at 30 °C. Following PBS-T washes, all rolling circle products (RCPs) were hybridised with LM (Cy5 labelling mix with DAPI) for 30 min at room temperature, the section was washed with PBS-T and dehydrated with 70% and 100% ethanol. The hybridisation chamber was removed and the slide mounted with SlowFade Gold Antifade Mountant (Thermo S36937). Imaging of Cy5-labelled RCPs at this stage acted as a QC step to confirm RCP ('anchor') generation and served to identify spots during decoding. Imaging was conducted using a Perkin Elmer Opera Phenix Plus High-Content Screening System in confocal mode with 1 μm z-step size, using a 63X (NA 1.15, 0.097 μm/pixel) water-immersion objective. Channels: DAPI (excitation 375 nm, emission 435-480 nm), Atto 425 (ex. 425 nm, em.

**Fig. 7 | Patterning of the fetal spinal cord in the axial planes. A** Spatial heatmaps of selected *HOXB* gene expression in the PCW9 cervical spinal cord. PCW, post-conception week. **B** Spatial heatmaps of spatially-variable *HOX* gene expression in the axial plane of the fetal spinal cord at PCW9 (upper row) and PCW7 (lower row). White asterisk indicates voxels with co-expression of motor column-defining *HOX* genes previously reported as mutually exclusive; *HOXC6* & *HOXC9* for cervical and thoracic samples, *HOXC9* & *HOXC10* for lumbar sample. Dot colour corresponds to expression of genes highlighted in the legend (reds- *HOXB*8/9, greens- *HOXC*5/6, blues-*HOXC*8/9, purples-*HOXC*10/*D*10). **C** Upper panel: In-situ sequencing data at PCW7 in the axial plane showing ventral cord cells that co-express motor column-defining *HOX* genes previously reported as mutually exclusive; *HOXC6* & *HOXC9* for cervical and thoracic samples, *HOXC9* & *HOXC10* for lumbar sample. Orange cells are co-expressing. Red boxes indicate zoomed regions for the lower panels. Lower Panel: High magnification in-situ sequencing images showing co-expression of HOX genes. Cells highlighted with a white outline are co-expressing. ISS, in-situ sequencing. Source data are provided as a Source Data file.

463–501 nm), Alexa Fluor 488 (ex. 488 nm, em. 500–550 nm), Cy3 (ex. 561 nm, em. 570–630 nm), Cy5 (ex. 640 nm, em. 650–760 nm).

Following imaging, each slide was de-coverslipped vertically in PBS (gently, with minimal agitation such that the coverslip 'fell' off to prevent damage to the tissue). The section was dehydrated with 70% and 100% ethanol, and a new hybridisation chamber secured to the slide. The previous cycle was stripped using 100% formamide (Thermo AM9342), which was applied fresh each minute for 5 min, then washed with PBS-T. Barcode labelling was conducted using two rounds of hybridisation, first an adapter probe pool (AP mixes AP1-AP6, in subsequent cycles), then a sequencing pool (SP mix, customised with Atto 425), each for 1 h at 37 °C with PBS-T washes in between and after. The section was dehydrated, the chamber removed, and the slide mounted and imaged as previously. This was repeated another five times to generate the full dataset of 7 cycles (anchor and 6 barcode bits).

### Image-based in situ sequencing (ISS) decoding
We employed the ISS decoding pipeline outlined in Li et al.[49]. This pipeline consists of five distinct steps. Firstly, we performed image stitching using Acapella scripts provided by Perkin Elmer, which generated two-dimensional maximum intensity projections of all channels for each cycle. Next, we employed Microaligner version 1.0.0[50] to register all cycles based on DAPI signals using the default parameters. For cell segmentation, we utilised a scalable algorithm that leverages CellPose version 2.0[51] as the segmentation method. The expected cell size is set to 70 pixels in diameter and further expanded 10 pixels to mimic the cytoplasm. To decode the RNA molecules, we employed the PoSTcode version 1.0 algorithm[52] with the following parameters: rna_spot_size=7, prob_threshold=0.95, trackpy_percentile=95, trackpy_separation=3. Furthermore, we assigned the decoded RNA molecules to segmented cells using STRtree and subsequently generated AnnData objects, using the AnnData version 0.11 python package[53].

### Multiplexed smFISH
Cryosections were processed using a Leica BOND RX to automate staining with the RNAscope Multiplex Fluorescent Reagent Kit v2 Assay (Advanced Cell Diagnostics, Bio-Techne), according to the manufacturers' instructions. Probes may be found in Supplementary Data 7. Prior to staining, fresh frozen sections were post-fixed in 4% paraformaldehyde in PBS for 15 min at 4 °C, then dehydrated through a series of 50%, 70%, 100%, and 100% ethanol, for 5 min each. Following manual pre-treatment, automated processing included digestion with Protease IV for 30 min prior to probe hybridisation. Tyramide signal amplification with Opal 520, Opal 570, and Opal 650 (Akoya Biosciences) and TSA-biotin (TSA Plus Biotin Kit, Perkin Elmer) and streptavidin-conjugated Atto 425 (Sigma Aldrich) was used to develop RNAscope probe channels. Stained sections were imaged as for ISS above.

### Single-cell RNA sequencing
The single-cell suspensions derived from each sample were loaded onto separate channels of a Chromium 10x Genomics single-cell 3'version 2 library chip as per manufacturer's protocol (10x Genomics; PN-120233). cDNA sequencing libraries were prepared as per the manufacturer's protocol and sequenced using an Illumina Hi-seq 4000 with 2x150bp paired-end reads.

### Alignment, quantification and quality control of single-cell data
Raw sequence reads in FASTQ format were processed and aligned to the GRCh38 version 1.2.0 human reference transcriptome through the Cellranger version v3.0.2 pipeline (10x Genomics) using default parameters.

The resulting expression matrices were processed with the SoupX package version 1.4.8 for R to estimate and remove cell-free mRNA contamination prior to analysis with the Seurat version 4.0.1 package for R[54,55].

Cells with fewer than 750 genes and greater than 10,000 genes were filtered, as well as those in which mitochondrial genes represented 10% or greater of total gene expression.

### Alignment, quantification and quality control of Visium
Raw FASTQ files and histology images were processed, aligned and quantified by sample using the Space Ranger software v.1.0.0. which uses STAR v.2.5.1b52 for genome alignment, against the Cell Ranger hg38 reference genome refdata-cellranger-GRCh38-3.0.2, available at: http://cf.10xgenomics.com/supp/cell-exp/refdata-cellranger-GRCh38-3.0.2.tar.gz.

Spots were automatically aligned to the paired H&E images by Space Ranger software. All spots under tissue detected by Space Ranger were included in downstream analysis.

### Doublet removal
Each 10x run was processed with the Scrublet version 0.2.2 pipeline using default parameters[55]. The louvain algorithm for clustering was then re-run at resolution 10 to provide an over-clustered manifold, and any clusters where greater than 20% of cells were called as doublets were removed. Subsequently, any remaining cells called as doublets were removed from analysis.

### Dimensional Reduction, clustering and annotation of single-cell data
Single-cell data were processed using the Seurat package v4.0.1 for R[56]. To account for variations in cell-cycle stage, Seurat's 'CellCycleScoring' function was performed on the remaining cells to produce a quantitative estimation of cell cycle stage. Log normalisation was then performed using the "NormalizeData" function with default parameters such that total number of counts per cell were normalized to 10,000 prior to data scaling, which used cell cycle score, mitochondrial gene expression level and the total unique molecular identifiers (UMIs) per cell as regression variables. Variable features were identified using the "FindVariableFeatures" function to select for 2000 most variably expressed genes. Principal-component (PC) analysis was then performed on log-transformed data using the "RunPCA" function, and the optimum number of PCs ($n = 67$) for downstream analysis was identified using molecular cross-validation (MCVR) in R[57]. The neighbourhood graph was then computed using these and other default parameters and the graph embedded in two dimensions using uniform manifold approximation and projection. Clustering of single-cell data

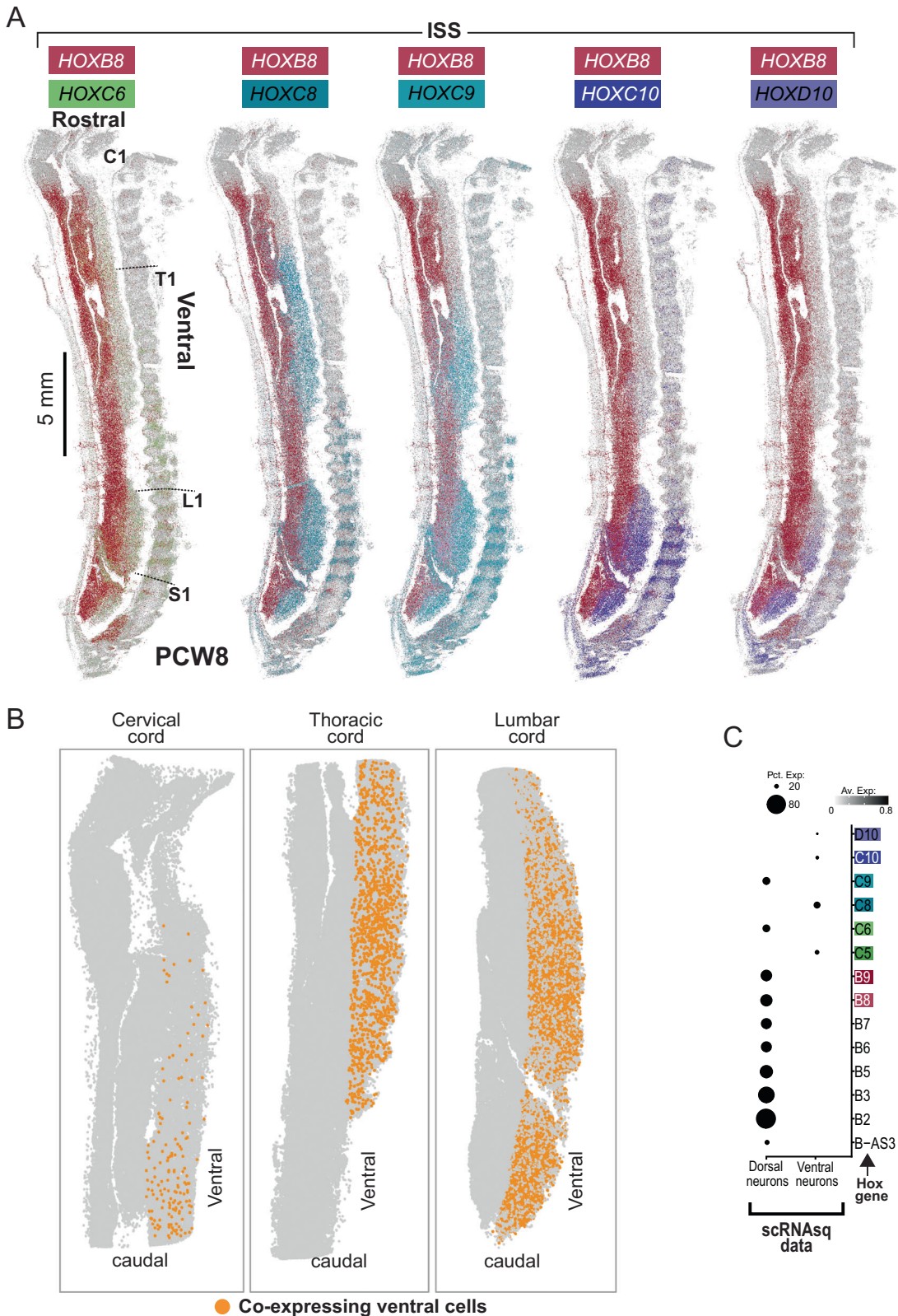

**Fig. 8 | Patterning of the fetal spinal cord in the sagittal plane. A** In-situ sequencing data showing expression of classical motor-column associated *HOX* genes in the sagittal plane, together with *HOXB8* expression. C1= first cervical vertebra, T1= first thoracic vertebra, L1= first lumbar vertebra, S1= first sacral vertebra. **B** In-situ sequencing data at post-conception week 8 in the sagittal plane showing ventral cord cells that co-express motor column-defining *HOX* genes previously reported as mutually exclusive; *HOXC6* & *HOXC9* for the cervical and thoracic cord, *HOXC9* & *HOXC10* for the lumbar cord. Orange cells are co-expressing. **C** Dotplot of mean expression values of *HOX* genes amongst single cells of the fetal cord, derived from single-cell RNA sequencing. Pct. Exp, percentage of expressing cells; Av. Exp, average expression in cell type.

has been performed by Louvain community detection on the neighbourhood graph with default resolution set to 1.

## Differential Gene Expression Analysis for single cell and visium data

Annotation of single cell clusters was performed by computing differentially expressed genes using the Wilcoxon rank sum test via Seurat's "FindMarkers" function, with genes requiring a minimum of 10% expression within a cluster to be returned, and a log2FC threshold of 0.1. Adjusted *p*-value (post bonferroni correction) cut off was set to 0.05. Cells were annotated based on differential expression of classical genes previously identified in literature (Supplementary Data 1). Regional (cervical, thoracic, lumbar, sacral) differential gene expression was performed using the same method; dermal and cord/neuroglial cells were excluded from this analysis, as almost all of the former derived from the sacral region of a single fetus and the latter from PCW5 and 7, which could not be reliably dissected into precise anatomical sections. To analyse expression in stationary and neural crest-derived cell types in the Visium data, we classified voxels based on whether their dominant imputed cell types was NCC-derived or not before plotting gene expression using Seurat's 'DotPlot' function and the Visutils version 0.2.1 R package's "plotVisiumMultyColours" function (for visium data).

For detecting genes that exhibit variable expression in the axial plane, spatially variable gene expression was computed on Visium data by applying the "FindSpatiallyVariableFeatures" function to regions of interest within the slide, using the top 3000 variable genes as input. Features were then extracted using the "SVFInfo" function.

Co-expression of motor column HOX genes in the ventral cord was then evaluated using the "WhichCells" function, specifying the combination of relevant HOX genes (HOXC6 & HOXC9 for the thoracic region; HOXC9 & HOXC10 for the lumbar region) requiring non-zero expression for a voxel to be annotated as co-expressing.

To explore gene expression that varied with rostro-caudal position, single-cell clusters (resting chondrocytes and perichondrium) were collapsed into pseudo bulk RNA data by summing the raw expression for each gene in the cluster in cervical and lumbar regions, at each time point. These data were then analysed using DESeq2, following the standard pre-processing workflow prior to constructing a negative binomial model using anatomical location and batch as input variables[58]. To elucidate genes that varied with location, a likelihood ratio test (LRT) was performed, with batch removed in the reduced model.

## Calculating per-cell HOX:SOX10 ratio

To calculate the ratio of lumbar HOX genes to SOX10 mRNA counts, per-cell log-normalised expression values of HOX genes expressed in NCC-derived cells were divided by the value for *SOX10*, producing a per-cell ratio value. Results were visualised using the ggplot2 version 3.3.3 package in R. Significance was calculated using the Wilcoxon rank-sum test, two-sided.

## Examining HOX regulons—DoRothEA

In order to test if genes that varied with rostro-caudal position were targets of HOX proteins, we accessed the R package DoRothEA version 3.19 regulon database, selecting confidence levels A-D; that is transcription factor targets with direct evidence of TF-target interaction. We then searched the database for the differentially expressed genes identified.

## Deconvolution of Visium data—cell2location

To map cell types identified by scRNA-seq in the profiled spatial transcriptomics slides, we used the cell2location version 0.1.3 method[17]. In brief, this involved first training a negative binomial regression model to estimate reference transcriptomic profiles for all the cell types identified with scRNA-seq. Next, lowly expressed genes are excluded as per recommendations for use of cell2location. Next, we estimated the abundance of each cell type in the spatial transcriptomics slides using the reference transcriptomic profiles of different cell types. Results were visualised using the Visutils package for R.

## Deconvolution of ISS data

To map cells types identified by scRNA-seq in the profiled ISS slides, we applied a nearest neighbour approach[59,60]. The single-cell data was then subsetted to include only genes in the ISS panel. An annoy index was next constructed from the single cell data using the annoy version 1.0.3 package for python, with the metric specified as euclidean[59]. This index was then used to calculate the K-nearest neighbour of each ISS cell in the single-cell dataset, and the identity of this neighbour applied to the ISS cell.

## Gene expression analysis of ISS data

ISS data was analysed as per single-cell RNA sequencing data, using the following workflow: AnnData objects were first converted for use with the Seurat version 4.0.1 package for R, using the "convertFormat" function of the sceasy R package[61]. Any cells expressing fewer than two-panel genes were removed, and the data normalised and scaled using the "NormalizeData" and "ScaleData" Seurat functions, respectively. Gene expression was visualised using the "FeaturePlot" Seurat function. To identify muscle cells, tendon cells and neurons using panel genes, the "WhichCells" function was used, specifying scaled expression of the marker genes MYOD1, TNMD and NTRK1 as >0.1 to identify each of these cell types respectively. To exclude any NTRK1-expressing mesenchymal cells, DRG cells were first isolated anatomically using the lasso function in the webatlas version 0.2.1 browser and their barcodes exported, prior to subsetting the Seurat objects for these cells[22]. To examine co-expression of motor column HOX genes in the ISS samples, the ventral half of the cord was selected with the lasso function, and the Seurat object subsetted based on these cell barcodes. Co-expression was then evaluated using the "WhichCells" function, specifying the combination of relevant HOX genes requiring non-zero expression for a cell to be annotated as co-expressing (HOXC6 & HOXC9 for the thoracic region; HOXC9 & HOXC10 for the lumbar region). HOX gene modules (Fig. 2) were computed using the "AddModuleScore" seurat function. Data were visualised using the "DotPlot" and "FeaturePlot" Seurat functions.

## Reporting summary

Further information on research design is available in the Nature Portfolio Reporting Summary linked to this article.

# Data availability

The new raw transcriptomic data generated in this study have been deposited in the EGA database under accession code EGAD00001009801 [https://ega-archive.org/studies/EGAS0000 1005090] and are accessible to researchers by application. The in situ sequencing data generated in this study have been deposited in the Bioimage archive database under accession code S-BIAD1117. Processed data can be downloaded and visualised at our data portal (https://spinal-development.cellgeni.sanger.ac.uk). Source data are provided with this paper.

# Code availability

No previously unreported custom computer code or algorithms were used to generate results that are reported in this paper.

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

## Acknowledgements
We thank Professor Denis Duboule and Professor James Briscoe for their invaluable advice for this study. This study was supported by the Wellcome Trust through institutional / programme grants and personal Fellowships (WT211276/Z/18/Z- institutional grant, 108413/A/15/D (S.B. & O.B.), 222902/Z/21/Z (S.B.) and 223135/Z/21/Z (J.E.L.).

## Author contributions
Study conception: J.E.L. and S.B. Data analysis and interpretation: J.E.L. with contributions from M.D.Y., S.B., O.B., N.D.A, P.M., S.A.T., A.F., T.L. and R.A.B. Experiments: K.R., E.T., J.E.L., P.B, I.U., A.P., L.M., L.B., L.R. and E.P. Tissue curation: X.H. Manuscript writing: J.E.L and S.B. with contributions from all authors. S.B. and O.B. co-directed the study.

## Competing interests
The authors declare no competing interests.
