## [Peer Review file · Nature Communications]

HOX gene expression in the developing human spine

Corresponding Author: Professor Sam Behjati

Version 0:

Reviewer comments:

Reviewer #1

(Remarks to the Author)

The authors did a lot of work to respond to my comments and other reviewer's comments, and they have addressed all my concerns.

One thing that I am still trying to understand is the expression of HOXB genes in most stationary tissues vs cord tissue. I understand that because of the different ages analyzed (PCW9-13 for most stationary tissue by single cell and PC7-9 for cord by spatial) it is not possible to directly compare the HOX codes between them, but I can't find a discussion about it. It seems like the dorsal cord follows yet a different HOX code than the rest of the tissues. Maybe it deserves 1 or 2 sentences in the discussion.

One more small question is regarding the nomenclature of the tissue ages. In the beginning it says "we collected 7 spines from fetuses aged between 5 and 13 weeks gestation. From post-conception week (PCW) 9 onwards (n=5)". Mentioning both "weeks of gestation" and "post-conception week" can be confusing, since these two terms are in principle different (with ~14 days difference between them).

Other than this, there are a few typos, so the manuscript would benefit from another round of proofreading. Just a few that I could find:

- Line 124-125: "We first aimed to identify HOX genes that reliably represented positional across all stationary cells, regardless of type" – missing the word information possibly?
- Line 153-154: "with HOXB6, HOXA5, HOXB3 and specific to the cervical region" – missing a another HOX gene after and
- Line 188-189: I believe it should say Fig. 4B, C (instead of Fig. 3B C)
- Figure 5 legend, replace second "D" with "E"
- Title for Extended Data Figure: change "maker" for "marker"

Reviewer #2

(Remarks to the Author)

The manuscript by Lawrence et al. combines several advanced single-cell and spatial omics techniques to study the development of the human fetal spine during the first trimester of pregnancy, culminating in a highly detailed developmental atlas. The study also investigates HOX gene expression along the rostrocaudal axis. Notably, it reveals that the progeny of neural crest lineage cells retain the anatomical HOX code of their origin. The authors extend their analysis to various organs, validating the HOX gene expression patterns. While the work provides a substantial dataset, its analysis primarily offers observational descriptions, limiting the full potential of the data and the manuscript.

Major concerns:

1. The analysis predominantly describes HOX gene expression across different areas of the ST sections or cells from varied rostrocaudal regions. The study lacks a detailed examination of how these HOX genes contribute to developmental processes such as cell proliferation, differentiation, or migration along the rostrocaudal or dorsal-ventral axis. The analysis relies heavily on gene plotting of HOX genes, providing limited novelty or deeper insight into potential mechanisms.
2. Some sections of the manuscript are descriptive without detailed analysis. For instance, lines 242-262 lack clarity on how the gene expression of BCL11A, LMXB, SALL3, ZIC1 (Fig 5B) relates to HOX genes. Are there interactions, cell type specificities, or are these genes co-expressed independently in the same area?

3. The cell type nomenclature requires more precision. In lines 259-262, the designation of HOXC5, 8, 10, and HOXD10 as exclusive to ventral neurons is uncertain. These areas might contain oligodendrocytes or migrating OPCs at this stage. The description of HOX gene expression in each cell type should be corroborated with additional markers in Visium and ISS.
4. The authors acknowledge the limitations in cell capture and in-depth analysis (line 334 onwards). However, the opportunity to utilize existing datasets to address these limitations was missed. An analysis of how HOX gene expression correlates with spatiotemporal gene expression is needed. Here are some papers that showed human spinal cord development across different regions: Li et al. (Nature Neuroscience 2023), integrated scRNAseq data with ST to delineate the development of cell type as well as specific gene expression for spatially distinct cell types. Such dataset and analysis could have been taken into consideration. Zhang et al. (EMBO Reports 2021), provided a large dataset covering first and second trimester with tissue dissected from different levels of the spinal cord. Other works, such as Andersen et al. (Nature Neuroscience 2023), and Rayon et al. (Development 2021), also provided late or early human spinal cord samples.
5. Although human samples are rare, it seems the number of cases for ST is very small. Would it be helpful by including existing datasets (listed but not limited above)?

Minor

6. In Fig 1B, the color code makes it difficult to distinguish cell types within the same lineage.
7. For Fig 3B, please maintain consistency in the dorsal and ventral positioning of the images.
8. Lines 242-245: Clarification is needed on what makes HOXB-AS3 "unique." It appears similar across all regions and is not expressed uniquely compared to other genes.

Reviewer #3

(Remarks to the Author)

I still have a major problem with the main conclusion of the manuscript, which asserts that migratory neural crest cells maintain their thoracic identity and gain additional HOX code based on the target tissue. To support this claim, the authors would need to demonstrate that during their initial specification and migration, these cells express one code, and then after settling down, this code is changed. However, I do not find evidence for this assertion in the manuscript. Therefore, a simpler explanation is that the neural crest expresses a rostro-caudal HOX code that differs from the rostro-caudal code of spinal cord tissue.

The manuscript has been significantly improved by the addition of in situ hybridization data. However, all the data are presented in a 'bird's-eye view,' lacking the cellular resolution that is the most useful feature of such data. In the absence of higher magnification images, it is difficult to establish the quality of the data, and how useful it will be. Specifically, the authors should demonstrate co-expression of HOX genes that they consider incongruent, either based on their analysis or on prior publications that have shown mutually exclusive patterns of HOX gene expression (e.g., Dasen & Jessell papers, which need to be discussed and cited).

I agree that the manuscript contains a substantial amount of data and could serve as a valuable resource. In that case, it is important to ensure that all processed and raw data are freely available to the scientific community. While the authors provide a link to an online platform for scRNA-seq data, similar access to spatial transcriptomics and in situ data seems to be missing. This would need to be rectified before publication. Below are additional specific comments.

1. The authors conclude that neural crest derivatives express a combination of thoracic source code and a gained positional code from the region to which they migrate. However, in the analysis of HOX genes in neural crest presented in ED4, it seems that cervical, thoracic, lumbar, and sacral neural crest express distinct HOX codes, but that the code differs from the code in stationary cells. Could this simply mean that different cell types and tissues utilize different HOX codes and there is no "universal" rostro-caudal HOX code that applies to all tissues? This interpretation would be consistent with known observations that proximo-distal mesoderm also relies on HOX code for patterning, but the code is not identical to the rostro-caudal code in the ventral spinal cord. As the authors point out and others have described, the rostro-caudal HOX code in ventral spinal cord is also very different from the one in the dorsal spinal cord. This interpretation would be simpler, more intuitive, and consistent with the above-mentioned observations and prior publications.
2. Furthermore, the in situ hybridization data with single-cell resolution demonstrating co-existence of the two codes is lacking (It is not clear what data are used in ED5B,C to call cells "co-expressing" - show example images of DRGs in high magnification with relevant HOX gene in situ and their quantification). This is a critical point, as another alternative explanation could be that the close apposition of neural crest cells with cells from other lineages in the target tissue confounds the low-resolution spatial transcriptomics data.
3. The pattern of HOX gene expression in the developing dorsal and ventral spinal cord has been described in detail by Dasen et al. in Cell 2005. It is not clear why the authors do not reference and discuss this work (particular focus should be on supplementary data showing the expression pattern of HOX genes along the rostro-caudal axis of the chick neural tube). This study demonstrated dorsal expression of HOXB cluster genes and differential expression of the rest of the HOX genes in the ventral spinal cord. This is a significant resource that would allow the authors to draw similarities between human and other vertebrate HOX patterns, at least in the neural tissue.
4. "The maintenance of lower level expression of other motor-column HOX genes, as evidenced by both spatial transcriptomics and in-situ sequencing, suggests that the cross-repression between these groups is not total at this point in development (PCW7-9) (Fig. 5C-E)." This data is not apparent from the referenced figure. It is necessary to evaluate and

demonstrate co-expression of individual HOX genes in the scRNA seq and in situ studies. This is particularly important in light of the above-mentioned paper that established strictly mutually exclusive patterns of HOX transcription factors in the chick and mouse ventral spinal cord. Do human HOX genes conform to the same rules? The difference in co-expression could be due to the timing differences as suggested by the authors, but additional possibilities are that this is a true species-specific difference, or that the observed overlap is true for RNA but not protein expression.

5. It would be helpful to also compare patterns of gene expression with the recent study by the Nedelec group that examined regulation and expression of HOX genes in human ventral spinal motor neurons in vitro (PMID: 33782043).

6. If this is primarily a resource paper, it will be important to share all the in situ and spatial transcriptomics data. The authors should add this data to their online browser or upload all the raw data to an accessible online database.

7. "In vertebrates, Hox genes are activated sequentially in a temporally-restricted manner, with 3' transcription commencing in response to Wnt signalling."

8. This statement is not accurate – the 3' Hox genes are primarily controlled by retinoic acid.

9. Panel 5E is mislabeled in the figure legend. It is important to demarcate cervical, thoracic, and lumbar boundaries as defined by the authors in this panel.

10. There are still numerous typos and grammatical errors throughout the manuscript. It needs to be carefully proofread.

Version 1:

Reviewer comments:

Reviewer #2

(Remarks to the Author)

The authors have addressed all my questions with detailed explanation. I support the publication for this manuscript.

Manuscript NCOMMS-24-00997-T
Lineage-specific HOX gene expression in the developing human spine
Response to Reviewers

Reviewer 1

#	Comment	Response
1.0	The authors did a lot of work to respond to my comments and other reviewer's comments, and they have addressed all my concerns	Thank you – we are glad the revised work answers your main questions
1.1	One thing that I am still trying to understand is the expression of HOXB genes in most stationary tissues vs cord tissue. I understand that because of the different ages analyzed (PCW9-13 for most stationary tissue by single cell and PC7-9 for cord by spatial) it is not possible to directly compare the HOX codes between them, but I cant find a discussion about it. It seems like the dorsal cord follows yet a different HOX code than the rest of the tissues. Maybe it deserves 1 or 2 sentences in the discussion.	Thank you for highlighting this finding. We have now expanded our discussion to discuss the pattern that seems to have emerged through this work and that of others. In animal models, the HOXB genes seem to be the first HOX genes expressed in the neural tube, in a non-specific manner. This is followed by a restricted expression pattern in the intermediate zone of the maturing neural tube. Our study suggests the next stage in this evolving expression pattern is a tight restriction to the dorsal horns, all the while continuing to lack collinearity. We have clarified this in the discussion and the results as per your suggestion. Changes to manuscript: 1) Lines 309-316: “These findings build on patterns of HOXB expression identified in earlier developmental stages in animals, whereby HOXB genes are initially ubiquitously expressed throughout the neural tube, then restricted to the intermediate zone, which traverses the dorsoventral axis of the neural tube^{31,32}. Analysis of a single publicly available PCW5 spatial transcriptomics axial section

suggested that at this earlier stage HOXB8 expression did not exhibit such clear dorsal restriction (**Extended Data Figure 7B**)³³. At PCW12, HOXB8 expression was very similar to PCW9 with clear restriction to the dorsal cord (**Extended Data Figure 7B**).”

2) Lines 415-427:

“Our study also investigated the expression of HOX genes along the dorsoventral axis of the spinal cord. Previous work with model organisms has uncovered the evolving role of the HOXB genes in neurogenesis. In the neural tube of the mouse and chick, HoxB genes are expressed before other Hox clusters, initially in a non-specific manner throughout the tube, before becoming somewhat restricted to the intermediate zone; an area bridging the ventricular zone (containing progenitors) and the mantle zone (containing post-mitotic cells)^{31,32}. Our study continued the study of this evolving expression pattern, showing with both single cell and spatial transcriptomics that at later time points (in the human) HOXB genes show restriction to the dorsal cord, and continue to lack any rostrocaudal collinearity. Perhaps after regulating neuronal delamination via LZTS1 at the intermediate zone in the neural tube, HOXB genes take on a further role in specifying sensory / dorsal identity in the cord. Further experiments in model systems examining the effect of HOXB inactivation at different developmental stages would shed light on their role at the stages studied here.”

1.2	One more small question is regarding the nomenclature of the tissue ages. In the beginning it says “we collected 7 spines from fetuses aged between 5 and 13 weeks gestation. From post-conception week (PCW) 9 onwards (n=5)”. Mentioning both “weeks of gestation” and “post-conception week” can be confusing, since these two terms are in principle different (with ~14 days difference between them).	Apologies for this – as you say these terms are not interchangeable. We have removed all mention of gestation in the paper for consistency.
1.3	Other than this, there are a few typos, so the manuscript would benefit from another round of proofreading. Just a few that I could find:  - Line 124-125: “We first aimed to identify HOX genes that reliably represented positional across all stationary cells, regardless of type” – missing the word information possibly? - Line 153-154: “with HOXB6, HOXA5, HOXB3 and specific to the cervical region” – missing a another HOX gene after and - Line 188-189: I believe it should say Fig. 4B, C (instead of Fig. 3B C) - Figure 5 legend, replace second “D” with “E” - Title for Extended Data Figure: change “maker” for “marker” 	Thank you for identifying these errors. We have now corrected these errors and ensured full independent proof reading of this final version.

Reviewer 2

#	Comment	Response
2.1	The manuscript by Lawrence et al. combines several advanced single-cell and spatial omics techniques to study the development of the human fetal spine during the first trimester of pregnancy, culminating in a highly detailed developmental atlas. The study also investigates HOX gene expression along the rostrocaudal axis. Notably, it reveals that the progeny of neural crest lineage cells retain the anatomical HOX code of their origin. The authors extend their analysis to various organs, validating the HOX gene expression patterns. While the work provides a substantial dataset, its analysis primarily offers observational descriptions, limiting the full potential of the data and the manuscript.	Thank you for the kind comments
2.2	The analysis predominantly describes HOX gene expression across different areas of the ST sections or cells from varied rostrocaudal regions. The study lacks a detailed examination of how these HOX genes contribute to developmental processes such as cell proliferation, differentiation, or migration along the rostrocaudal or dorsal-ventral axis. The analysis relies heavily on gene plotting of HOX genes, providing limited novelty or deeper insight into potential mechanisms	Thank you for these comments. Whilst the Reviewer poses an interesting set of questions, they are quite separate from what we aimed to investigate in this study and therefore cannot readily be answered using the methods available to us at our institute. The purpose of our work was to study the pattern of HOX gene expression through development in different lineages along different embryonic axes, not to study the cellular function of HOX genes during development or the mechanisms by which they effect these functions. If one wanted to pursue this, it would require an entirely different set of experiments, such as perturbing HOX genes in an animal model and observing the effects on (for example) the organisation of the spinal cord or the dorsal root ganglia.

		We would therefore politely suggest that this is genuinely beyond the scope of our work.
2.3	Some sections of the manuscript are descriptive without detailed analysis. For instance, lines 242-262 lack clarity on how the gene expression of BCL11A, LMXB, SALL3, ZIC1 (Fig 5B) relates to HOX genes. Are there interactions, cell type specificities, or are these genes co-expressed independently in the same area?	We apologise if this section caused confusion. The intention of using those four classical sensory neuron TFs was to orient the reader to the location of sensory neurons at that stage of development in the axial plane. However, we have now removed these plots to avoid any confusion. Changes to manuscript: 1) Lines 307-309: Removed reference to sensory neuron transcription factors. “Whilst group B genes were all dorsally expressed, their precise expression patterns varied within this domain (Fig. 5A).” 2) Figure 5 (appended to this document) Removed panel B
2.4	The cell type nomenclature requires more precision. In lines 259-262, the designation of HOXC5, 8, 10, and HOXD10 as exclusive to ventral neurons is uncertain. These areas might contain oligodendrocytes or migrating OPCs at this stage. The description of HOX gene expression in each cell type should be corroborated with additional markers in Visium and ISS.	We apologise for the lack of clarity here. The data referred to in these lines is single cell RNA sequencing, rather than Visium (we of course agree with the reviewer that a voxel can contain many cells – see the discussion). We have now made this clearer in the figure by making the label above the plot larger, and moving. Changes to manuscript: 1) Figure 5 Altered dotplot of scRNAseq markers to make source of the data clearer

2) Lines 332-338:

“We also examined expression of these genes by region in the single cell data (almost all of which originated from week 5 & 7 samples), again finding HOXB genes were exclusive to dorsal neurons in the developing cord with a high percentage of cells expressing them (Fig. 5E). The ventral HOX genes showed mixed expression (Fig5E). HOXC5, HOXC8, HOXC10 & HOXD10 were exclusive to ventral neurons, expressed by small percentages of the total cells (presumably reflecting their restriction along the rostro-caudal axis to particular regions), but HOXC6 and HOXC9 were more dorsally expressed (Fig5E).”

2.5

The authors acknowledge the limitations in cell capture and in-depth analysis (line 334 onwards). However, the opportunity to utilize existing datasets to address these limitations was missed. An analysis of how HOX gene

Thank you for these suggestions to increase the amount of data we could use to investigate HOX expression along the rostrocaudal axis.

expression correlates with spatiotemporal gene expression is needed. Here are some papers showed human spinal cord development across different regions: Li et al. (Nature Neuroscience 2023), integrated scRNAseq data with ST to delineate the development of cell type as well as specific gene expression for spatially distinct cell types. Such dataset and analysis could have been taken into consideration. Zhang et al. (EMBO Reports 2021), provided a large dataset covering first and second trimester with tissue dissected from different levels of the spinal cord. Other works, such as Andersen et al. (Nature Neuroscience 2023), and Rayon et al. (Development 2021), also provided late or early human spinal cord samples.

The Reviewer highlights the fact that we were unable to investigate HOX expression along the rostrocaudal axis in neuroglial cells (line 334 onwards) as these were (as one would expect) only captured in the earliest samples, which we could not dissect reliably into regions.

The papers / datasets the Reviewer highlights fall into three categories:

- 1) Single cell sequencing only (Rayon et al)
- 2) A combination of single nuclear and single cell sequencing (Zhang et al; Anderson et al)
- 3) A combination of single cell and spatial transcriptomics (Li et al)

Unfortunately, all these studies dissected the cord away from other tissues, thus have no utility in investigating the expression of HOX genes across all lineages of the spine- the main aim of our work. We do not believe they would contribute to our question (with the exception of two ST samples from Li et al which we have now incorporated) for reasons outlined below:

1) **Rayon dataset** – This single cell dataset isolated the neural tube/nascent cord (mostly prior to our sampling window) from all other tissues and did not dissect by region. This data, therefore, cannot be used to investigate HOX expression along the rostrocaudal axis. Furthermore, because we do not have ST or ISS data from this time period, we could not contextualise cells from their dataset using our label transfer methods. Integrating this data with ours would simply increase the size of our single cell “atlas” dataset to include more early neuroprogenitors. Whilst useful to the scientific community, this task has already been performed by Anderson et al (see below). Therefore, the amount of work required

to achieve this would seem to have little benefit to the scientific community or our main question.

2) **Zhang dataset** - Again this dataset only includes the cord, so lacks the breadth of tissue sampling required to investigate lineage-specific HOX expression, which is the main aim of our study. Furthermore, only GA10 (PCW8) cord samples onwards were dissected by region (see their table EV1) ie comparable to our study, therefore this study would not help in investigating how neuroglial cells express HOX genes along the rostrocaudal axis. Much of the data is single-nuclear which would pose great technical challenges for integration. Their earliest [whole cord] sample is GA7 (PCW5); the same as our study. There is no spatial data.

3) **Li dataset** – Again, this dataset only includes the cord and did not dissect by anatomical region. We have however used their PCW6 and PCW12 data to show HOXB expression in the dorsal cord (see **Extended Data Figure 7**). The use of their ISS dataset would be of limited benefit as our 123-gene panel was specifically designed to answer the question of HOX expression along the AP and DV axes. Their panel does not contain the same gene set as it was designed for a different purpose (primarily celltype mapping). Finally, we would argue that our ST and ISS experiments better preserved tissue architecture by maintaining the cord in its context (vertebral column and associated tissues plus DRGs).

4) **Andersen dataset** - This dataset includes samples from PCW 15 and 16, but again only included the spinal cord in their dissection and were not dissected by anatomical region. This again prevents any conclusions about the use of HOX genes across lineages and axes in the developing cord. Perhaps most importantly, this paper has already integrated the data from Rayon and Zhang (see figure 7

of their paper) to form a very rich single cell ‘atlas’, which would seem to reduce the novelty and utility of us performing that same labour-intensive task here, as suggested by the Reviewer.

Changes to manuscript:

1) Extended Data Figure 7:

Added new panel showing PCW5 and PCW12 HOXB8 expression from public data (Appended to this document)

2) Lines xx-xx

*“Analysis of a single publicly available PCW5 spatial transcriptomics axial section suggested that at this earlier stage HOXB8 expression did not exhibit such clear dorsal restriction (**Extended Data Figure 7B**)³³. At PCW12, HOXB8 expression was very similar to PCW9 with clear restriction to the dorsal cord (**Extended Data Figure 7B**).”*

2.6	Although human samples are rare, it seems the number of cases for ST is very small. Would it be helpful by including existing datasets (listed but not limited above)?	Please see 2.5
2.7	In Fig 1B, the color code makes it difficult to distinguish cell types within the same lineage.	Thank you for this feedback. We have now changed the figure to include a wide range of colours (figure appended to this document) B 2.8	For Fig 3B, please maintain consistency in the dorsal and ventral positioning of the images.	Thank you for this suggestion – we have now rotated the panel in question by 180 degrees, and matched the orientation with panel C. Figure appended to this document.
2.9	Lines 242-245: Clarification is needed on what makes HOXB-AS3 “unique.” It appears similar across all regions and is not expressed uniquely compared to other genes.	Thank you for highlighting this – this was a poor choice of words. We have now reworded the manuscript. Changes to manuscript:

		1) Lines 305-307: “Intriguingly, this included the antisense transcript HOXB-AS3, which was expressed in the medial aspect of the dorsal horn and the periaqueductal region (Fig. 5A; Extended Data Figure 7A).”
--	--	---

Reviewer 3

#	Comment	Response
3.1	I still have a major problem with the main conclusion of the manuscript, which asserts that migratory neural crest cells maintain their thoracic identity and gain additional HOX code based on the target tissue. To support this claim, the authors would need to demonstrate that during their initial specification and migration, these cells express one code, and then after settling down, this code is changed. However, I do not find evidence for this assertion in the manuscript. Therefore, a simpler explanation is that the neural crest expresses a rostro-caudal HOX code that differs from the rostro-caudal code of spinal cord tissue	Thank you for raising this very valid point. In order to explore this, we compared NCC-derived cells in the lumbar samples at PCW9 and PCW 12, which represented our “best” samples in terms of numbers of NCC-derived cells captured. We reasoned that if the NCC-derived cells do indeed gain local gene expression as they mature (ie the expression is dynamic, not fixed), then cells at PCW12 should express more local HOX gene and less SOX10 than those at PCW9. We did find this to be the case for two of the three relevant genes. However, we are careful to caveat the limitations of this analysis in the text, including the discussion, and have carefully reworded the text throughout to be more agnostic on this point of retention / adoption of different HOX codes, removing references to code retention of an expression pattern. Changes to Manuscript 1) Extended Data Figure 5F (appended to this document)

2) Lines 246-262:

“The expression of a region-specific set of HOX genes in NCC-derived cells may commence before their migration begins, during their migration, or be adopted after they arrive in their new location. Our data may provide some clues in this regard by assessing, in individual cells, the correlation of the expression of the classical NCC marker SOX10 versus local HOX code genes. If, for example, we found that more immature cells (i.e. those with higher expression of SOX10) expressed local HOX code genes less strongly, it may suggest that at least some of the local HOX code expression is acquired after migratory neural crest settled in their new location and identity. To pursue this analysis, we compared this ratio of SOX10 versus local HOX code expression of the most immature and mature cells that we captured in sufficient numbers, i.e. lumbar cells at PCW9 and PCW12 (with the local HOX code HOXC9, HOXA9, and HOXC10). Comparing PCW9 to PCW12 lumbar cells, we found a significant

		difference in the ratio for two HOX genes (HOXA9 and HOXC10); more immature cells expressed relatively more SOX10 than local HOX code (Extended Data Figure 5F). Although this observation would be consistent with a dynamic acquisition of the local HOX code upon arrival of neural crest cells, ultimately this question can only be answered through lineage tracing experiments. As these cannot be performed in human tissue, it would have to be pursued in model systems.” 3) Lines 463-468: “Similarly, when investigating the dynamics of HOX expression in NCC-derived cells, we were limited to two samples (lumbar PCW9 & 12) as no other two samples from different fetuses at the same anatomical level captured sufficient NCC-derived cells to allow adequately powered analysis. Furthermore, PCW9 is relatively advanced in terms of neural crest specification; studying earlier time points would shed further light on HOX expression dynamics.”
3.2	The manuscript has been significantly improved by the addition of in situ hybridization data. However, all the data are presented in a 'bird's-eye view,' lacking the cellular resolution that is the most useful feature of such data. In the absence of higher magnification images, it is difficult to establish the quality of the data, and how useful it will be. Specifically, the authors should demonstrate co-expression of HOX genes that they consider incongruent, either based on their analysis or on prior publications that have shown	Thank you for raising this concern. May we perhaps clarify what may be a misunderstanding of the methods. As you say, the in situ sequencing data are single cell resolution. For visualisation we do indeed provide a “bird’s eye view”. For statistical analyses, however, we quantify the data at the resolutiton of single cells and molecules. For example, we identified muscle and tendon cells through canonical marker expression, then isolated the DRG histologically

mutually exclusive patterns of HOX gene expression (e.g., Dasen & Jessell papers, which need to be discussed and cited).

using the lasso function using the WebAtlas browser software. This then enabled investigation of neuronal HOX expression (as congruent – “local code” or incongruent – “source code” or co-expression). We went on to classify individual neurons in terms of their HOX expression (non-expressing, local code only, source code only or co-expressing; see **Extended Data Figure 5B-E**). Furthermore, in response to 3.6 we have now leveraged the single cell resolution of ISS to show individual cord cells co-express mutually exclusive HOX genes.

All of this analysis was performed with ISS data using the Seurat package for R, in effect treating it as single cell data with a reduced number of genes in the expression matrix (that is, at single-cell resolution as the Reviewer has requested here).

In the methods and text we have now tried to clarify how we isolated the DRG and then treated those cells as we would a single cell dataset using the same computational workflow. Apologies if this was not clear. We have now included a methods schematic in **Extended Data Figure 5B** to make it so.

Changes to Manuscript

1) Lines 230-234:

“For single-cell resolution in-situ sequencing data, we again showed the same expression pattern through analysis of NTRK1-expressing neurons within the dorsal root ganglia (DRG), which was first isolated anatomically using the WebAtlas browser’s lasso tool, before subsetting for NTRK1-positive cells only (Fig. 4B & C; Extended Data Figure 5B; See Methods)”

2) Lines 806-823 (methods):

“ISS data was analysed as per single-cell RNA sequencing data, using the following workflow: AnnData objects were first converted for use with the Seurat version 4.0.1 package for R, using the “convertFormat” function of the sceasy R package⁶¹. Any cells expressing fewer than two panel genes were removed, and the data normalised and scaled using the “NormalizeData” and “ScaleData” Seurat functions respectively. Gene expression was visualised using the “FeaturePlot” Seurat function. To identify muscle cells, tendon cells and neurons using panel genes, the “WhichCells” function was used, specifying scaled expression of the marker genes MYOD1, TNMD and NTRK1 as >0.1 to identify each of these cell types respectively. To exclude any NTRK1-expressing mesenchymal cells, DRG cells were first isolated anatomically using the lasso function in the webatlas browser and their barcodes exported, prior to subsetting the Seurat objects for these cells²². To examine co-expression of motor column HOX genes in the ISS samples, the ventral half of the cord was selected with the lasso function, and the Seurat object subsetted based on these cell barcodes. Co-expression was then evaluated using the “WhichCells” function, specifying the combination of relevant HOX genes requiring non-zero expression for a cell to be annotated as co-expressing (HOXC6 & HOXC9 for the thoracic region; HOXC9 & HOXC10 for the lumbar region). HOX gene modules (Fig. 2) were computed using the “AddModuleScore”

		seurat function. Data were visualised using the “DotPlot” and “FeaturePlot” Seurat functions.” 3) Extended Data Figure 5B (appended to this document): 3.3	I agree that the manuscript contains a substantial amount of data and could serve as a valuable resource. In that case, it is important to ensure that all processed and raw data are freely available to the scientific community. While the authors provide a link to an online platform for scRNA-seq data, similar access to spatial transcriptomics and in situ data seems to be missing. This would need to be rectified before publication. Below are additional specific comments.	Thank you. The Visium data is already available through our online portal together with the scRNAseq data, and should have been available to the Reviewer from the initial submission onwards. We apologise if there was difficulty in accessing this. Please could you contact the Editor if this remains an issue. We can confirm that at the time of submitting this document, we were able to access the data from outside our confines. The spatial transcriptomics and scRNAseq data has been uploaded in raw format on EGA as detailed in the manuscript. We are uploading ISS data to Bioimage archive (accession number pending).
3.4	The authors conclude that neural crest derivatives express a combination of thoracic source code and a gained positional code from the region to which they migrate. However, in the analysis of HOX genes in neural crest presented in ED4, it seems that cervical, thoracic, lumbar, and sacral neural crest express	Thank you for raising this important point. We have now extensively altered the text wording throughout to be more agnostic on this point, removing references to a “retention” of a code. We have now also stated the fact that neural crest derivatives seem to express their own combination of region-specific

distinct HOX codes, but that the code differs from the code in stationary cells. Could this simply mean that different cell types and tissues utilize different HOX codes and there is no “universal” rostro-caudal HOX code that applies to all tissues? This interpretation would be consistent with known observations that proximo-distal mesoderm also relies on HOX code for patterning, but the code is not identical to the rostro-caudal code in the ventral spinal cord. As the authors point out and others have described, the rostro-caudal HOX code in ventral spinal cord is also very different from the one in the dorsal spinal cord. This interpretation would be simpler, more intuitive, and consistent with the above-mentioned observations and prior publications.

HOX genes (which we list) in addition to the universal expression of thoracic HOX.

As per your suggestion, we include this in the discussion in the context of differing expression patterns in the ventral cord, and the limb bud.

Changes to Manuscript

1) Lines 36-38:

“Interestingly, derivatives of the migratory neural crest lineage expressed a HOX code corresponding to their point of origin in the thorax whilst simultaneously expressing a set of region-specific HOX genes at destination.”

2) Lines 47-52:

“We built a detailed developmental atlas to examine the expression of *HOX* genes across different cell types along the antero-posterior axis, distilling a broad *HOX* code applicable to mesenchyme-derived anatomically fixed cell types. We found that unlike these stationary cells, derivatives of neural crest cells expressed the anatomical *HOX* code corresponding to their point of origin within the neural crest, whilst also expressing a set of region-specific *HOX* genes at their end destination.”

3) Lines 92-94:

“We show varying HOX gene expression across cell types through development, highlighting that cells derived from the migratory neural crest express HOX genes that correspond to their point of origin along the rostrocaudal axis.”

4) Lines 213-224:

*“In addition, these cells also expressed some HOX genes expressed by co-located stationary mesenchyme-derived cells. This included HOXB2 in NCC-derived cells in the cervical region and HOXC9, A9 & C10 to the lumbar region, together with HOXD11 in the sacral region (**Fig. 4A; Table S3**). However, NCC-derived cells also expressed HOX genes not differentially expressed by local mesenchyme-derived cells, including HOXC4 in cervical cells, HOXC8 in lumbar cells and HOXA9 & C10 in sacral cells (**Fig. 4A; Table S3**). In other words, these cells exhibited thoracic HOX expression regardless of their final destination (perhaps consistent with their truncal neural crest origin), whilst also expressing their own local HOX code, which partially overlapped with the expression pattern identified in stationary, mesenchyme-derived cells. This idiosyncratic HOX expression pattern seemed to remain consistent across the developmental stages sampled (**Extended Data Figure 4**).”*

5) Lines 388-403

“Our work distilled from orthogonal methods and multiple independent specimens a core HOX code in mesenchyme-

		derived tissues in the human spine between the 9th and 13th weeks post conception. Interestingly, we found that during this developmental window, NCC-derived cells seemed to express a thoracic HOX code, perhaps in keeping with their truncal origin within the neural crest, whilst also expressing a set of region-specific HOX genes at their end destination. Whilst there was some overlap between these region-specific genes and the anatomical code derived from mesenchyme-derived cells, differences were also evident. This suggests that NCC-derived cells utilise a different HOX code to mesenchyme for positional information. This idiosyncratic expression pattern is in keeping with other human tissues, such as the mesenchyme of the limb bud (which has its own HOX expression patterns) and the ventral and dorsal cord as discussed here⁴⁷. We showed this pattern of HOX expression corresponding to neural crest origin held true in the fetal limb and adrenal gland (containing truncal neural crest-derived cells), and the fetal gut (containing vagal neural crest-derived cells). Together, these findings shed further light on the utilisation of this enigmatic set of genes by different tissues during development.”
3.5	Furthermore, the in situ hybridization data with single-cell resolution demonstrating co-existence of the two codes is lacking (It is not clear what data are used in ED5B,C to call cells “co-expressing” - show example images of DRGs in high magnification with relevant HOX gene in situs and their quantification). This is a critical point, as another alternative explanation could be that the close apposition of neural crest cells with cells from other lineages in the target tissue confounds the low-resolution spatial transcriptomics data.	Apologies for the lack of clarity – this is not from visium data but from ISS data. We have now tried to be clearer in the text that these figure panels (now C-E) represent ISS data. Apologies if this was not clear previously. We have also made it clear that these plots (and the associated quantification the reviewer refers to) are derived from ISS data through the schematic drawing discussed in 3.2 and by adding labels to the plots stating “ISS data”, and through tweaking the text.

We have also added zoom boxes to the RNA-ISH as per your helpful suggestion.

Changes to Manuscript

1) Extended Data Figure 5A(appended to this document)

2) Extended Data Figure 5C-E (labels):

3) Lines 230-234:

“For single-cell resolution in-situ sequencing data, we again showed the same expression pattern through analysis of *NTRK1*-expressing neurons within the dorsal root ganglia (DRG), which was first isolated anatomically using the WebAtlas browser’s lasso tool, before subsetting for *NTRK1*-positive cells only (Fig. 4B & C; Extended Data Figure 5B; See Methods)”

4) Lines 806-823 (methods):

“ISS data was analysed as per single-cell RNA sequencing data, using the following workflow: AnnData objects were first

converted for use with the Seurat version 4.0.1 package for R, using the “convertFormat” function of the sceasy R package⁶¹. Any cells expressing fewer than two panel genes were removed, and the data normalised and scaled using the “NormalizeData” and “ScaleData” Seurat functions respectively. Gene expression was visualised using the “FeaturePlot” Seurat function. To identify muscle cells, tendon cells and neurons using panel genes, the “WhichCells” function was used, specifying scaled expression of the marker genes MYOD1, TNMD and NTRK1 as >0.1 to identify each of these cell types respectively. To exclude any NTRK1-expressing mesenchymal cells, DRG cells were first isolated anatomically using the lasso function in the webatlas browser and their barcodes exported, prior to subsetting the Seurat objects for these cells²². To examine co-expression of motor column HOX genes in the ISS samples, the ventral half of the cord was selected with the lasso function, and the Seurat object subsetted based on these cell barcodes. Co-expression was then evaluated using the “WhichCells” function, specifying the combination of relevant HOX genes requiring non-zero expression for a cell to be annotated as co-expressing (HOXC6 & HOXC9 for the thoracic region; HOXC9 & HOXC10 for the lumbar region). HOX gene modules (Fig. 2) were computed using the “AddModuleScore” seurat function. Data were visualised using the “DotPlot” and “FeaturePlot” Seurat functions.”

3.6	“The maintenance of lower level expression of other motor-column HOX genes, as evidenced by both spatial transcriptomics and in-situ sequencing, suggests that the cross-repression between these groups is not total at this point in development (PCW7-9) (Fig. 5C-E).” This data is not apparent from the referenced figure. It is necessary to evaluate and demonstrate co-expression of individual HOX genes in the scRNA seq and in situ studies. This is particularly important in light of the above-mentioned paper that established strictly mutually exclusive patterns of HOX transcription factors in the chick and mouse ventral spinal cord. Do human HOX genes conform to the same rules? The difference in co-expression could be due to the timing differences as suggested by the authors, but additional possibilities are that this is a true species-specific difference, or that the observed overlap is true for RNA but not protein expression.	Thank you. Unfortunately, due to the fact that cord neurons in the single cell data were captured from the early samples, which were not reliably dissectible by region, we are unable to analyse HOX co-expression in ventral neurons as a function of position along the rostro-caudal axis using the single cell data. Fortunately the ISS data is most helpful in answering this question and maintains the single cell resolution. We have now performed co-expression analysis on the ISS data to demonstrate the lack of mutual exclusivity at the single cell level. We have expanded the discussion to include the Reviewer’s helpful points about the possible reasons for the lack of mutual exclusivity identified in our experiments. Changes to Manuscript 1) Figure 5B & C (appended to this document)
------------	---	---

2) Extended Data Figure 7C (appended to this document)

3) Lines 351-360

*“Whilst spatial transcriptomics data confirmed that HOXC6 expression dominates the ventral cervical cord, and group 10 genes dominate the ventral lumbar cord, both datasets revealed the surprising maintenance of lower level expression of other motor-column HOX genes in these regions (**Fig. 5B-D**). In the axial ST sections, at PCW7 & 9, multiple voxels covering the ventral cervical and thoracic cord coexpressed HOXC6 and HOXC9, and several voxels covering the ventral lumbar cord coexpressed HOXC9 and HOXC10/D10 (**Fig. 5B**, white asterisks; see methods). Further analysing this at single-cell resolution using ISS at PCW7 (from sections 12µm adjacent to the visium samples), we confirmed that some ventral cells at each level did indeed express, in combination, motor column-defining HOX genes*

previously reported as mutually exclusive (Fig 5C; see methods).”

4) Lines 429-444

“A further interesting finding in our study was the co-expression of certain motor-pool determining HOX genes in the ventral cord which had previously been reported as mutually exclusive in animal models. As discussed above regarding HOXB gene expression, this may again be a reflection of evolving patterns of HOX expression during development; our study window being comparatively late compared to the animal models. However it remains possible that this is a species-specific feature. A previous in vitro study by Mouilleau et al used the timing of retinoic acid (RA) application to culture to recapitulate the temporal nature of motor neuron development; cervical neurons develop earlier than lumbar neurons³⁴. Adding RA at day 3 onwards resulted in strong HOXC6 expression, whereas addition at day 6 onwards resulted in a clear shift towards HOXC9, recapitulating expression along the rostro-caudal axis³⁴. However, addition at day 5 did seem to result in some overlap in the expression of these two genes, though co-localisation analysis of these two supposedly exclusive genes was not performed. Further studies of their expression dynamics using similar methods would be enlightening. Finally, whilst RNA assays provide high resolution and sensitivity, they do not necessarily reflect the presence of protein; it may be that at the protein level HOX distribution in the human fetal cord is identical to that of animal models.”

3.7	It would be helpful to also compare patterns of gene expression with the recent study by the Nedelec group that examined regulation and expression of HOX genes in human ventral spinal motor neurons in vitro (PMID: 33782043).	Thank you for highlighting this excellent work. We have now incorporated it into our manuscript as highlighted below Changes to Manuscript 1) lines 318-324: “The identity taken up by ventral progenitors in the cord is thought to be highly dependent on HOX expression. This was recently highlighted in vitro by Mouilleau et al, who showed that stalling FGF-induced temporal activation of HOX genes in human pluripotent stem cell (hPSC)-derived progenitors resulted in cervical motor neuron types, whereas accelerating HOX progression produced lumbar neuronal subtypes³⁴. We therefore aimed to explore these trends in vivo by examining HOX expression in the fetal ventral spinal cord along the rostro-caudal axis.” 2) lines 433-441: “However it remains possible that this is a species-specific feature. A previous in vitro study by Mouilleau et al used the timing of retinoic acid (RA) application to culture to recapitulate the temporal nature of motor neuron development; cervical neurons develop earlier than lumbar neurons³⁴. Adding RA at day 3 onwards resulted in strong HOXC6 expression, whereas addition at day 6 onwards resulted in a clear shift towards HOXC9, recapitulating expression along the rostro-caudal axis³⁴. However, addition

		at day 5 did seem to result in some overlap in the expression of these two genes, though co-localisation analysis of these two supposedly exclusive genes was not performed."
3.8	If this is primarily a resource paper, it will be important to share all the in situ and spatial transcriptomics data. The authors should add this data to their online browser or upload all the raw data to an accessible online database.	Thank you. The Visium data is already available through our online portal together with the scRNAseq data, and has been uploaded in raw format on EGA as detailed in the manuscript. We are uploading ISS data to Bioimage archive (accession number pending).
3.9	"In vertebrates, Hox genes are activated sequentially in a temporally-restricted manner, with 3' transcription commencing in response to Wnt signalling." - This statement is not accurate – the 3' Hox genes are primarily controlled by retinoic acid.	Thank you for raising this interesting point. This statement was based off PMIDs: 37322110, 27633012 28728680. However we acknowledge the complexity of this mechanism and the remaining uncertainty regarding the initiation of 3'HOX transcription, and have therefore removed this statement from the text. Changes to manuscript 1) Lines 70-72: "In vertebrates, Hox genes are activated sequentially in a temporally-restricted manner, beginning with 3' transcription, with the timing of progression through the cluster governed by CTCF binding sites"
3.10	Panel 5E is mislabeled in the figure legend. It is important to demarcate cervical, thoracic, and lumbar boundaries as defined by the authors in this panel.	Apologies for this error. We have corrected the legend and added the labels you suggest. Thank you.

Figure 1

Figure 3

Figure 5

Extended Data Figure 5

Extended Data Figure 7